# Wasserstein Modality Alignment Makes Your Multimodal Transformer More Robust

**Zhuo Zhi**                                                                *zhuo.zhi.21@ucl.ac.uk*
*Department of Electronic and Electrical Engineering*
*University College London* *

**Yuxuan Sun**                                                             *yuxuan.sun.22@ucl.ac.uk*
*Department of Electronic and Electrical Engineering*
*University College London*

**Qiangqiang Wu**                                                        *qiangqwu2@cityu.edu.hk*
*Department of Computer Science*
*City University of Hong Kong*

**Ziquan Liu**                                                                *ziquan.liu@qmul.ac.uk*
*School of Electronic Engineering and Computer Science*
*Queen Mary University of London* *

**Miguel Rodrigues**                                                      *m.rodrigues@ucl.ac.uk*
*Department of Electronic and Electrical Engineering*
*University College London*

**Reviewed on OpenReview:** *https://openreview.net/forum?id=2IkaUZdB62*

## Abstract

Multimodal fusion with a multimodal transformer is an effective method for both early and late fusion paradigms. However, in a multimodal transformer, the modality fusion is performed solely through the self-attention mechanism, which is originally designed for unimodal token sequences. To improve the self-attention mechanism for handling multimodal input, a parametric adapter model, like the Q-former in BLIP-2, is often used to align tokens from different modalities. Our empirical study unveils that only using the self-attention layer to perform the modality fusion makes the model less robust to missing modalities and input noise, as the model will overly rely on one certain modality. To improve the robustness of the transformer, our paper proposes an implicit approach based on Wasserstein distance that aligns tokens from different modalities without using any additional trainable parameters. Our empirical study shows that the implicit modality alignment improves the effectiveness of the multimodal Transformer in discriminative tasks, as well as its robustness to input noise and missing modalities. We conduct experiments on four downstream task datasets, including 2-modalities and 3-modalities tasks. We also consider different fusion paradigms, i.e., early and late fusion. The experimental results show that our proposed method has a significant improvement in both performance and robustness over all baselines across all datasets and fusion paradigms.

## 1 Introduction

Multimodal machine learning (MML) mimics human perception by integrating multiple modalities such as text, audio, images, video, and sensor data to form a comprehensive understanding of the world. Many

---

*Corresponding author

multimodal models have been applied to various tasks like multimodal medical diagnostics Hayat et al. (2022a), sentiment analysis Zadeh et al. (2018) and malicious speech detection Kiela et al. (2020).

Aligning heterogeneous data in multimodal learning is crucial since such data often exhibit distinct distributions, representations, and noise levels. Proper alignment enhances the uniform representation of these diverse data types, leading to improved performance and robustness in multimodal tasks Ghahremani Boozandani & Wachinger (2024); Liang et al. (2024); Kim et al. (2020). To achieve better modality alignment, various strategies are applied in large-scale multimodal models, such as the Q-Former in BLIP-2 Li et al. (2023), contrastive learning in CLIP Radford et al. (2021) and Imagebind Girdhar et al. (2023).

Multimodal fusion based on a one-tower transformer, named as multimodal transformer (MT), by its flexibility and simplicity, are widely used for a variety of multimodal learning tasks Lee et al. (2023); Nagrani et al. (2021); Zhi et al. (2024); Ma et al. (2021). Although the multimodal transformer can handle multimodal tokens as the input due to the flexibility of self-attention layers, it lacks a mechanism for modality alignment during the fine-tuning process. In other words, it is not optimal to rely on a pre-trained multimodal transformer to achieve the modal alignment Kim et al. (2021); Wang et al. (2021). Taking ViLT Kim et al. (2021) as an example, tokens from different modalities are concatenated together and processed by the multimodal transformer. Several learning tasks, such as image text matching and word patch alignment are applied during the pre-training phase to ensure the modality alignment. However, the alignment process is not enforced in the fine-tuning stage if the input tokens are from multimodal sources, which leads to deteriorated performance in a downstream task.

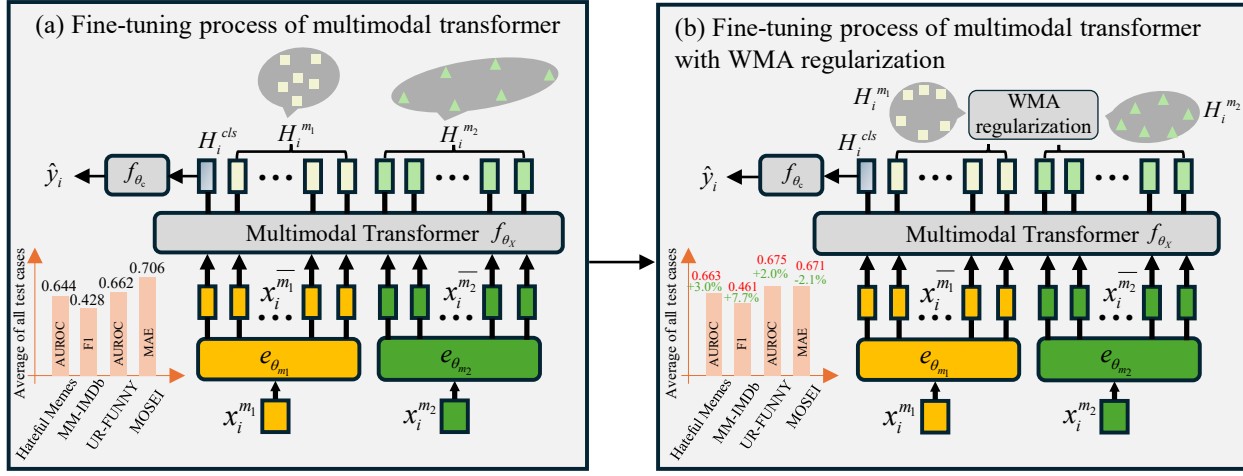

Figure 1: The overview of the proposed method (using early fusion structures as an example). (**a**) Fine-tuning the multimodal transformer on the downstream task. The image data $x_i^{m_1}$ and text data $x_i^{m_2}$ are firstly processed by the embedding operation $e_{\theta_{m_1}}$ and $e_{\theta_{m_2}}$ (in late fusion structure, $e_{\theta_{m_1}}$ and $e_{\theta_{m_2}}$ refer to the feature extraction by the pretrained encoders, i.e., Bert for text and ViT for image. ) to get the token sequence $x_i^{\overline{m}_1}$ and $x_i^{\overline{m}_2}$, which then are concatenated and processed by the multimodal transformer $f_{\theta_X}$. $H_i^{cls}$ in the multimodal transformer is input into the classifier $f_{\theta_c}$ for predicting the label $\hat{y}_i$. Note that there is no specific module to adjust modality alignment in (a). (**b**) We propose the WMA regularization method to search for the optimal modality alignment for the target task. WMA calculates the OT distance between $H_i^{m_1}$ and $H_i^{m_2}$ to represent the alignment degree of two modalities. The target OT distance range is set to achieve the optimal alignment. Our proposed WMA method effectively improves both the performance and robustness of the model, as shown in the lower left corner of (**1**) and (**2**) ( 'Average of all test cases' refers to the average result of normal test, test with noise and the test with missing modality. See more details in Table. 1 and Table. 2)

.

To address this issue, we propose Wasserstein Modality Alignment (WMA), an implicit regularization method to align the Wasserstein distance between two modalities within a multimodal transformer, as shown in Fig. 1. To make the computation feasible, we use the popular optimal transport (OT) distance Peyré et al. (2019) as the instantiation of Wasserstein distance. We regularize the degree of modality alignment by adjusting the OT distance between different modalities' feature distributions. Interestingly, our empirical study demonstrates that directly minimizing the OT distance between two modalities often leads to inferior performance, so our WMA aligns two modalities with a task-dependent modality distance. In the practical sense, the proposed WMA is a plug-and-play method and does not introduce any additional trainable parameters. Our main contributions are three-fold:

- We propose to perform the modality alignment in the fine-tuning process of a pre-trained multimodal transformer without any additional adapter and design the Wasserstein Modality Alignment based on the optimal transport distance to achieve lightweight modality alignment in the feature tokens of the transformer.

- Instead of minimizing the OT distance between any two modalities, our WMA is a task-dependent modality alignment method that can handle different requirements for the degree of modality alignment.

- We evaluate our proposed WMA with three strong baselines on four datasets, including 2-modalities and 3-modalities tasks. We also consider different fusion paradigms, including both early fusion and late fusion. Our experimental results demonstrate significant improvements in performance and especially robustness across all tasks and test cases. In early fusion paradigm, the averaged improvement of our method compared with the best result from all baselines over all datasets are 1.4%, 1.5%, 1.4%, 0.7% and 1.4%, respectively. For the late fusion paradigm, the improvement are 1.5%, 1.8%, 1.3%, 2.3% and 1.9%.

The paper is organized as follows. Sec. 2 gives an overview of related research. Sec. 3 introduces the proposed method. Sec. 4 shows the experiment results and the analysis. Finally, Sec. 5 summarizes the paper, its limitations, and future work.

## 2 Related Work

**Modality alignment in multimodal learning.** Almost all large-scale multimodal models use specific strategies for modal alignment. Contrastive loss is a popular approach that promotes related modality alignment by boosting the similarity of positive sample pairs Li et al. (2021); Radford et al. (2021); Girdhar et al. (2023). LLaVA trains an image-to-text adapter to align the two modalities Liu et al. (2024). Flamingo achieves modality alignment by combining a pretrained vision encoder and a language model through a series of gated cross-attention layers, allowing for effective interaction between visual and textual inputs Alayrac et al. (2022). In Li et al. (2023), BLIP-2 employs a lightweight Querying Transformer to connect frozen image encoders with large language models for modality alignment. In contrast, the multimodal transformer, i.e., ViLT Kim et al. (2021) and SimVLMWang et al. (2021) only employ the agent task such as image text matching and word patch alignment for aligning modality during pre-training, lacking such approach at fine-tuning stage. To solve this problem, we propose an implicit regularization method to adjust the alignment of the multimodal transformer during the fine-tuning process.

**Robustness of multimodal learning.** Modality noise and absence are two challenges to the robustness of multimodal learning. In ML for healthcare, the patient may be missing the data such as an X-ray due to economic/timing issues Zhi et al. (2024). In addition, some sensor data may be accompanied by a lot of noise due to improper wear. A similar situation occurs in vision-language tasks. For example, some online recommender models are unable to receive images uploaded by users or receive blurry images with a lot of noise due to network issues. Sijie et al. Mai et al. (2022) propose the multimodal information bottleneck to filter out noisy information in unimodal representations. Md et al. Islam & Iqbal (2022) apply a cooperative multitask learning-based guided multimodal fusion approach to get robust performance on noisy and misaligned sensor data. For the missing modality problem, Ma et al. (2021) reconstructs the

missing modalities using modality priors and Bayesian Meta-Learning during the inference phase. Lee et al. (2023) propose the missing-aware prompts to learn the patterns of complete and incomplete samples. In Zhi et al. (2024), an approach inspired by in-context learning is proposed to improve the data efficiency for multimodal learning under missing modality and data scarcity. However, these methods require additional parameters to enhance the incomplete samples. Differently, we employ the nonparametric regularization approach to obtain robust multimodal representation.

## 3 Proposed Method

We first describe the problem definition, our motivation and the proposed method is elaborated on later.

### 3.1 Problem setting

We consider the multimodal transfer learning problem with a downstream dataset $\mathcal{D}$ containing multimodal input samples. For notation simplicity, we assume there are two modalities in the dataset, i.e., $\mathcal{D} = \{x_i^{m_1}, x_i^{m_2}, y_i\}_{i=1}^N$ where $y_i$ is the label. Note that our framework can handle any number of modalities in principle and We will describe how to extend it to 3-modalities tasks later. When we employ a pretrained multimodal transformer for solving the target task, some embedding operations or feature extraction are performed firstly performed on the input data $x_i^{m_1}$ and $x_i^{m_2}$:

$$x_i^{\bar{m}_1} = e_{\theta_{m_1}}(x_i^{m_1}) = [m_{1cls}; m_{11}; \cdots; m_{1L_{m_1}}], \tag{1}$$

$$x_i^{\bar{m}_2} = e_{\theta_{m_2}}(x_i^{m_2}) = [m_{2cls}; m_{21}; \cdots; m_{2L_{m_2}}], \tag{2}$$

where $e_{\theta_{m_1}}$ and $e_{\theta_{m_2}}$ refer to the embedding operation for two modalities such as linear projection, position embedding and modality type embedding Kim et al. (2021) in early fusion structure. In late fusion structure, $e_{\theta_{m_1}}$ and $e_{\theta_{m_2}}$ refer to the feature extraction by the pretrained encoders, i.e., Bert for text and ViT for image. $m_{1cls}$ and $m_{2cls}$ are the added classification head token and $L_{m_1}$ and $L_{m_2}$ are the number of embedded tokens/feature. $[;]$ means the concatenate operation. Then, the multimodal transformer $f_{\theta_X}$ inference the output tokens $H_i$ by

$$H_i = f_{\theta_X}([x_i^{\bar{m}_1}; x_i^{\bar{m}_2}]) = [H_i{}^{cls}; H_i{}^{m_1}; H_i{}^{m_2}], \tag{3}$$

where $H_i{}^{m_1} \in \mathbb{R}^{L_{m_1} \times d}$ and $H_i{}^{m_2} \in \mathbb{R}^{L_{m_2} \times d}$ are the processed features for two modalities, $d$ is the embedding dimension. $H_i{}^{cls} \in \mathbb{R}^{1 \times d}$ is the final classification head token which can be input into an added classifier/regressor $f_{\theta_c}$ for predicting the label and minimizing the loss:

$$\hat{y}_i = f_{\theta_c}(H_i{}^{cls}), \ell_{task}^{(i)} = \ell_{cls}(\hat{y}_i, y_i), \tag{4}$$

where $\ell_{cls}$ is the task-dependent loss function such as cross-entropy and $\ell_{task}^{(i)}$ is the loss value for the $i$th sample.

The issue with the fine-tuning process described above is that it lacks the approach for aligning $x_i^{m_1}$ and $x_i^{m_2}$ or their representation in this multimodal transformer. We will introduce our proposed method for solving this problem later.

### 3.2 Motivation

Although the strength of multimodal learning over unimodal learning has been proved both in theory Huang et al. (2021); Lu (2023) and in practice Radford et al. (2021), it is plagued by missing modalities Zhi et al. (2024); Lee et al. (2023) and input noise Papandreou et al. (2009). One of the reasons for the brittleness of multimodal learning is that it heavily relies on one certain modality to make the prediction as a result of the distributional gap between different modalities. In other words, some modalities are more easily by the training process because those modalities are more learnable than others. The phenomenon is relevant to shortcut learning Geirhos et al. (2020) where the model will use the shortcut features to complete the given learning task and fail to generalize to out-of-distribution scenarios. To illustrate the phenomenon

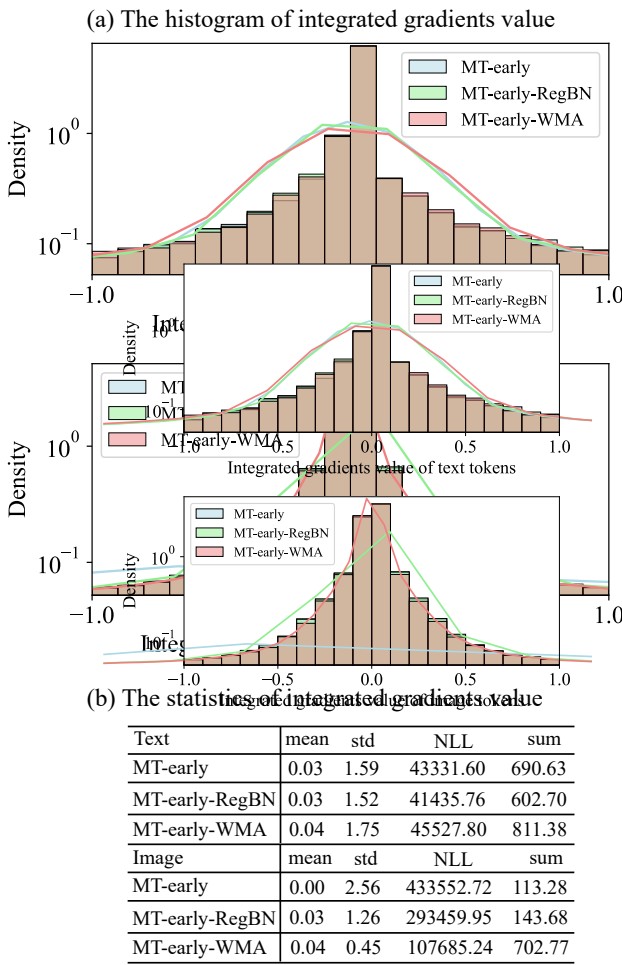

(a) The histogram of integrated gradients value

(b) The statistics of integrated gradients value

| Text | mean | std | NLL | sum |
|---|---|---|---|---|
| MT-early | 0.03 | 1.59 | 43331.60 | 690.63 |
| MT-early-RegBN | 0.03 | 1.52 | 41435.76 | 602.70 |
| MT-early-WMA | 0.04 | 1.75 | 45527.80 | 811.38 |
| Image | mean | std | NLL | sum |
| MT-early | 0.00 | 2.56 | 433552.72 | 113.28 |
| MT-early-RegBN | 0.03 | 1.26 | 293459.95 | 143.68 |
| MT-early-WMA | 0.04 | 0.45 | 107685.24 | 702.77 |

Figure 2: The Integrated gradients value of input token under Hateful Memes dataset and early fusion structure. We calculate the Integrated gradients value for every input token for all samples and put them together to obtain the distribution analysis. The bigger Integrated gradients value refers to the bigger contribution to the final classification. (**a**) The histogram of the Integrated gradients values. (**b**) The statistics of the distribution of the Integrated gradients values. MT-early, MT-eary-RegBN and MT-early-WMA refer to the original MT, MT with RegBN regularization Ghahremani Boozandani & Wachinger (2024) and MT with our proposed WMA regularization. The performance and robustness of three methods can be find in Table 1. From the statistics, we find that MT-early heavy relies on text for decision due to the sum of the Integrated gradients values of the text token 690.63 is much higher than that of the image token 113.28. The RegBN method alleviates the problem slightly with the numbers 602.70 and 143.68. Our proposed WMA method solves this problem – the contribution of text token and image token is almost identical–811.38 and 702.77. In WMA, the image modality has a more Gaussian-like attribution value distribution compared with others, meaning that the model exploits the weak modality to learn the task.

in multimodal learning, we use the explanation tool Integrated Gradients Sundararajan et al. (2017) to analyze a fine-tuned multimodal transformer on a two-modality dataset Hateful Memes under the early fusion structure, as shown in Fig. 2. Integrated Gradients is an attribution method that computes the contribution of each input feature to the output of a neural network by integrating gradients along a path from a baseline input to the actual input. With this tool, we obtain the contribution of each input token to the final classification.

It is observed from Fig. 2 that standard fine-tuning mainly uses the strong modality (text) to make the prediction while the weak modality (image) is not sufficiently exploited. To address the observed shortcut learning issue and make the model more robust to novel test environments, we propose to control the distributional gap between different modalities. As the distribution of easily learned modalities is regularized to be close to other modalities, the model is forced to avoid shortcut learning and use all modalities to learn the task.

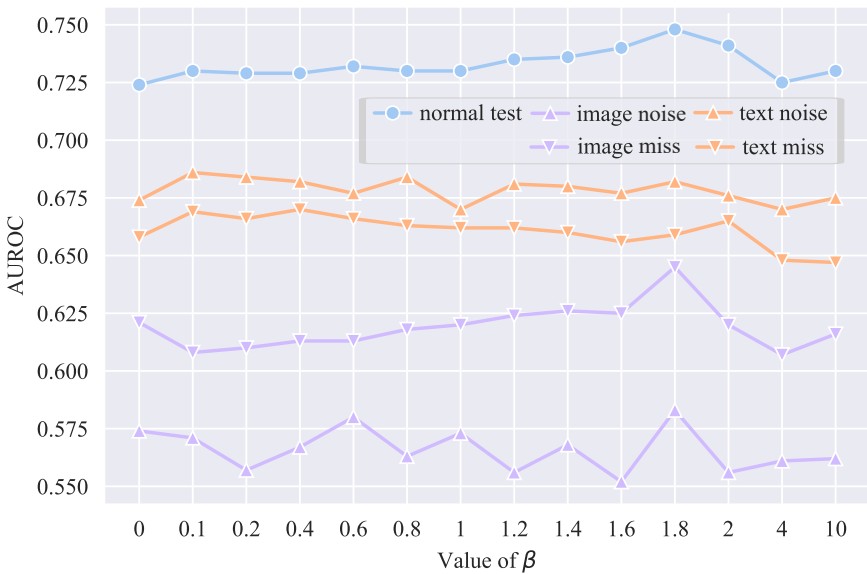

Figure 3: The performance of MT-early-WMA on Hateful Memes datasets in early fusion structure under $\alpha = 1$ with different value of $\beta$. When $\beta = 0$, it equals to minimize the OT distance. The best overall performance is gained at $\beta = 1.8$. The performance and robustness of the model are not exactly positively/negatively correlated with the value of $\beta$, indicating that the alignment of the model should be task-dependent, instead of minimizing the OT value. See more results in Table 5-8.

## 3.3 Wasserstein Modality Alignment

We propose the Wasserstein Modality Alignment (WMA), an implicit regularization for adjusting the alignment of different modalities in the multimodal transformer. For keeping computation efficient, we use the OT distance as the instantiation of Wasserstein distance and represent the alignment degree of two modalities by it. Uniquely, WMA searches the task-dependent optimal alignment through two hyperparameters rather than directly minimizing the OT distance.

For the feature $H_i^{m_1}$ and $H_i^{m_2}$, the optimal transport problem is defined as

$$W(H_i^{m_1}, H_i^{m_2}) = \min_{T \in \Sigma(\sigma, \delta)} \langle \mathbf{C}, T \rangle, \tag{5}$$

where $\mathbf{C} \in \mathbb{R}^{L_{m_1} \times L_{m_2}}$ is a manually defined cost matrix, with each element $c_{pq}$ representing the distance between the $p$th token of $H_i^{m_1}$ and the $q$th token of $H_i^{m_2}$. The optimal solution $T^\star$ is known as the optimal

transport plan. The set $\Sigma(\sigma, \delta)$ is defined as:

$$\Sigma(\sigma, \delta) = \left\{ T \in \mathbb{R}_+^{L_{m_1} \times L_{m_2}} \mid T\mathbf{1}_{L_{m_2}} = \sigma, \ T^\top \mathbf{1}_{L_{m_1}} = \delta \right\}, \tag{6}$$

where $\sigma$ and $\delta$ are the normalized distributions for $H_i^{m_1}$ and $H_i^{m_2}$, respectively, which are given by:

$$\sigma = \frac{1}{L_{m_1}} \mathbf{1}_{L_{m_1}}, \quad \delta = \frac{1}{L_{m_2}} \mathbf{1}_{L_{m_2}}, \tag{7}$$

where $\mathbf{1}_{L_{m_1}}$ and $\mathbf{1}_{L_{m_2}}$ are vectors of ones with lengths $L_{m_1}$ and $L_{m_2}$, respectively. To keep the computation efficient, we apply the IPOT algorithm Xie et al. (2018) to solve this problem.

The optimal transport cost $D_{m1m2}^i$ is calculated as

$$D_{m_1,m_2}^i = \langle \mathbf{C}, T^* \rangle. \tag{8}$$

We use $D_{m_1,m_2}^i$ as the reference for the alignment degree and manually set the target value to regularize the model parameters during the fine-tuning phase by modifying the loss function

$$\ell_{task}^{(i)} = \ell_{cls}(\hat{y}_i, y_i) + \alpha(D_{m_1,m_2}^i - \bar{D}_{m_1,m_2})^2. \tag{9}$$

We use the average $\bar{D}_{m_1,m_2}$ of the first batch at model initialization (by applying the pretrained weight) as a basis and set the search range by a combination of hyperparameters $\alpha$ and $\beta$. By this strategy, we do not minimize $D_{m_1,m_2}^i$, instead, we search for optimal alignment for the different modalities in the fine-tuning phase. Fig. 3 shows the search results for the Hateful Memes dataset in early fusion structure under $\alpha = 1$ with different values of $\beta$. The best overall performance is achieved at $\beta = 1.8$ which demonstrates the superiority of the proposed WMA over minimizing OT values. For 3-modalities tasks $H_i = [H_i^{cls}; H_i^{m_1}; H_i^{m_2}; H_i^{m_3}]$, we can easily modify Eq. 9 to

$$\begin{aligned}
\ell_{task}^{(i)} = \ell_{cls}(\hat{y}_i, y_i) + \alpha((D_{m_1,m_2}^i - \beta\bar{D}_{m_1,m_2})^2 \\
+ (D_{m_1,m_3}^i - \beta\bar{D}_{m_1,m_3})^2 \\
+ (D_{m_2,m_3}^i - \beta\bar{D}_{m_2,m_3})^2).
\end{aligned} \tag{10}$$

## 4 Experiment

We first introduce the experimental settings and then present the experimental results of our methods and baselines on five datasets and two fusion paradigms, demonstrating the effectiveness of our method.

### 4.1 Experimental Setting

**Datasets.** We select two 2-modalities datasets and two 3-modalities datasets across different downstream tasks to evaluate our proposed method.

- Hateful Memes Kiela et al. (2020). This is a binary classification task with two modalities, image and text. The task is to detect the maliciousness of memes. The numbers of the samples in the training/val/testing dataset are 8500, 500 and 1000.

- MM-IMDb Arevalo et al. (2017). This is a multi-label (25 labels) classification task with two modalities, image and text. The task is to tag the film. The numbers of the samples in the training/val/testing dataset are 32278, 5411 and 16120.

- UR-FUNNY Hasan et al. (2019). This is a binary classification task with three modalities, text, video and audio. The task is to detect humor in talks. The number of samples in the training/val/testing dataset are 8074, 1034, 1058.

- MOSEI Zadeh et al. (2018). This is a regression task with three modalities, text, video and audio. The task is to recognize the degree of sentiment. The number of samples in the training/val/testing dataset are 16265, 1869, 4643.

- MedFuse-I Hayat et al. (2022b). This is a real-world dataset which contains EHR and X-ray data for each patient. This binary classification task aims to predict in-hospital mortality after the first 48 hours spent in the ICU. The numbers of the samples in the training/val/testing dataset are 18845, 2138 and 5243.

**Metrics.** We set the metrics for each dataset according to the tasks. For Hateful Memes, UR-FUNNY and Medfuse-I, we use the AUROC as evaluation metrics. For MM-IMDb and MOSEI, we use F1 score and MAE, respectively.

**Multimodal Transformer for different fusion paradigms.** We consider two popular multimodal fusion paradigms by multimodal transformer: early fusion and late fusion. For early fusion, we use a 12-layer and 12-head transformer and initialize it with the pretrained weight ViLT-B Kim et al. (2021). For late fusion, we employ a 2-layer and 8-head transformer and randomly initialize it. All input tokens are added with modality embedding and a cls token by following Kim et al. (2021).

**Input data processing.** In early fusion structure, for vision-language tasks Hateful Memes and MM-IMDb, we follow the operation in ViLT Kim et al. (2021): the text is embedded by Bert and the image is split into patches (same with ViT). For UR-FUNNY and MOSEI, we use MultiBench Liang et al. (2021) to get the embedded feature of three modalities. For Medfuse-I, we use a linear layer to embed the EHR data and the same way with ViLT to process the X-ray image. In late fusion structure, for vision-language tasks Hateful Memes and MM-IMDb, a ViT-base and a Bert-base are employed to extract the feature of two modalities, while for UR-FUNNY and MOSEI, three GRU models are involved for feature extraction by following Liang et al. (2021). For Medfuse-I, we follow the operation in Hayat et al. (2022b).

**Baseline.** We select the following strong baseline for comparison. These methods focus on efficiently adjusting modal fusion to improve performance in multimodal learning.

- Standard transfer learning. We add a classifier/regressor for the target task and fine-tune all layers of the models, which is denoted as MT-early and MT-late in the latter part.

- RegBN Ghahremani Boozandani & Wachinger (2024). Batch normalization of multimodal data with regularization (RegBN) uses the Frobenius norm as a regularizer term to address the side effects of confounders and underlying dependencies among different modality sources. RegBN provides a plug-and-play approach to early and late fusion.

- GWMAC Gong et al. (2022). Gromov-Wasserstein multi-modal alignment and clustering (GWMAC) learns the GromovWasserstein barycenter of their kernel matrices. GW barycenter is associated with the GW distances between the different modalities to the clusters, and the optimal transport plans corresponding to the GW distances help to achieve the alignment and the clustering of the multimodal data jointly. This method only provides a late fusion version, and we extend it to an early fusion framework.

- MIB Mai et al. (2022). The multimodal information bottleneck (MIB) learns the minimal sufficient representation for a given task by maximizing the mutual information between the representation and the target and simultaneously constraining the mutual information between the representation and the input data. The method is also compatible with early fusion and late fusion frameworks.

**Hyperparameter settings.** We set the batch size for Hateful Memes, MM-IMDb, UR-FUNNY and MOSEI as 128, 64, 256, 256. The learning rate search range is [1e-3, 5e-4, 1e-4, 5e-5, 1e-5]. The learning rate strategy is linear decay with warm-up. The search range of $\alpha$ is set as [0.1, 0.2, 1.0, 5.0]. The search range of $\beta$ is [0.1, 0,2, 0.4, 0.6, 0.8, 1.0, 1.2, 1.4, 1.6, 1.8, 2.0, 4.0, 10.0]. Early stopping with patience 5 is applied for selecting the weight. Experiments are running on Tesla V100 GPUs.

**Robustness test setting.** We test the robustness of the model in two ways: test with modality noise and test with missing modality. When simulating noise in a particular modality, we consider three kinds

of noise for image data: Gaussian noise, salt and pepper noise and Poisson noise. For text data, we use swap adjacent letters, randomly permute middle section, keyboard typos, sticky keys and omission to create the noisy input. For time series data (i.e., some features from MultiBench Liang et al. (2021) and EHR data), we add the Gaussian Noise, uniform noise and Poisson noise. When simulating a missing modality, we randomly remove a proportion of the samples for that modality by setting all pixels to 0 for the image, replacing the text with an empty string and setting all values to 0 for time series. For Gaussian noise, the noise level is set to the standard deviation of the noise. For the salt and pepper noise, the noise level refers to the proportion of total pixels perturbed. For the uniform noise, the noise level refers to the noise range. For the text data noise, the noise level is the probability of randomly applying noise to a word. We set the noise level to be 0.25, 0.5, 0.75 to the image data, text data and time series data. The final result is the average of the performance at different noise levels. For the missing modality test, We set 25%, 50% and 75% as the missing proportion and calculate the average performance. Note that the real-world dataset Medfuse-I lacks X-ray images for 74% of samples and we do not manually set additional missing cases to simulate real application scenarios.

Table 1: Results of our proposed method with the baseline on all datasets and test cases for early fusion. The bold numbers mean the best performance. The bigger AUROC and F1 and smaller MAE refer to better performance.

| Datasets | Metric | Methods | normal test | test with modality noise | | | test with missing modality | | | |
| | | | normal | image/video noise | text/EHR noise | audio noise | image/video missing | text missing | audio missing | average |
|---|---|---|---|---|---|---|---|---|---|---|
| Hateful Memes | AUROC↑ | MT-early | 0.730 | 0.547 | 0.666 | - | 0.626 | 0.653 | - | 0.644 |
| | | MT-early-GWMAC | 0.732 | 0.554 | 0.680 | - | 0.626 | **0.669** | - | 0.654 |
| | | MT-early-RegBN | 0.736 | 0.546 | 0.680 | - | 0.620 | 0.650 | - | 0.646 |
| | | MT-early-MIB | 0.743 | 0.560 | 0.661 | - | 0.619 | 0.648 | - | 0.646 |
| | | MT-early-WMA | **0.748** | **0.583** | **0.682** | - | **0.645** | 0.659 | - | **0.663** |
| MM-IMDb | F1↑ | MT-early | 0.551 | 0.251 | 0.462 | - | 0.464 | 0.412 | - | 0.428 |
| | | MT-early-GWMAC | 0.553 | 0.353 | 0.470 | - | 0.472 | 0.422 | - | 0.454 |
| | | MT-early-RegBN | 0.558 | 0.216 | 0.463 | - | 0.451 | 0.429 | - | 0.423 |
| | | MT-early-MIB | **0.563** | 0.209 | 0.454 | - | 0.473 | 0.423 | - | 0.424 |
| | | MT-early-WMA | **0.563** | **0.355** | **0.471** | - | **0.477** | **0.437** | - | **0.461** |
| UR-FUNNY | AUROC↑ | MT-early | 0.700 | 0.645 | 0.678 | 0.635 | 0.612 | 0.670 | 0.696 | 0.662 |
| | | MT-early-GWMAC | 0.705 | 0.656 | 0.678 | 0.631 | 0.618 | 0.673 | 0.701 | 0.666 |
| | | MT-early-RegBN | **0.714** | 0.613 | 0.665 | 0.616 | 0.618 | 0.645 | 0.692 | 0.652 |
| | | MT-early-MIB | 0.711 | 0.641 | 0.682 | 0.63 | 0.606 | 0.670 | 0.690 | 0.666 |
| | | MT-early-WMA | 0.712 | **0.681** | **0.686** | **0.643** | **0.624** | **0.674** | **0.703** | **0.675** |
| MOSEI | MAE↓ | MT-early | 0.633 | 0.691 | 0.834 | 0.672 | 0.651 | 0.826 | 0.635 | 0.706 |
| | | MT-early-GWMAC | 0.624 | 0.679 | 0.827 | **0.657** | 0.632 | 0.817 | 0.634 | 0.696 |
| | | MT-early-RegBN | 0.617 | 0.714 | 0.877 | 0.665 | 0.622 | 0.848 | 0.629 | 0.704 |
| | | MT-early-MIB | 0.619 | 0.679 | 0.852 | 0.666 | 0.628 | 0.831 | 0.629 | 0.701 |
| | | MT-early-WMA | **0.615** | **0.676** | **0.824** | 0.664 | **0.619** | **0.813** | **0.623** | **0.691** |
| Medfuse-I | AUROC↑ | MT-early | 0.840 | 0.762 | 0.723 | - | - | - | - | 0.763 |
| | | MT-early-GWMAC | 0.844 | 0.787 | 0.749 | - | - | - | - | 0.793 |
| | | MT-early-RegBN | 0.855 | 0.751 | 0.722 | - | - | - | - | 0.776 |
| | | MT-early-MIB | 0.852 | 0.744 | 0.715 | - | - | - | - | 0.770 |
| | | MT-early-WMA | **0.861** | **0.796** | **0.755** | - | - | - | - | **0.804** |

Table 1 and 2 present the quantitative results of our proposed method WMA with the baselines across all the datasets and test cases under early and late fusion, respectively. From Table 1 and 2, we summarize the following observations:

- Our proposed WMA is more performant than the MT in all datasets and test cases under both early and late fusion paradigms. Specifically, in the average performance of all test cases, the improvements are 3.0%, 7.7%, 2.0%, 2.1% 5.4% for five datasets in early fusion and 2.4%, 2.8%, 1.8%, 3.6%, 3.4% in late fusion. Note that WMA in early fusion benefits the model more than in late fusion, indicating that the distributional regularization is an essential component to make the self-attention-based modality fusion more robust. Our proposed WMA method is also competitive compared to the best results from all baselines. To be specific, MT-early-WMA surpasses the best result at 1.4%,1.5%,1.4%, 0.7% and 1.4% in five datasets. For MT-late-WMA, the number comes to 1.5%, 1.8%, 1.3%, 2.3% and 1.9%.

- GWMAC exhibits a similar trend as the proposed WMA method in most datasets, which shows improvement in both normal test and robustness compared with the MT, but not as much as WMA. What's worse, GWMAC introduces more parameters and computing burden. It's worth noting that MIB and RegBN show the opposite trend with GWMAC and WMA, i.e., in most of the datasets, the performance of the normal tests is improved (some of them even achieve the highest score),

Table 2: Results of our proposed method with the baseline on all datasets and test cases for late fusion. The bold numbers mean the best performance. The bigger AUROC and F1 and smaller MAE refer to better performance.

| Datasets | Metric | Methods | normal test | test with modality noise | | | test with missing modality | | | |
|---|---|---|---|---|---|---|---|---|---|---|
| | | | normal | image/video noise | text/EHR noise | audio noise | image/video missing | text missing | audio missing | average |
| Hateful Memes | AUROC↑ | MT-late | 0.718 | 0.652 | 0.666 | - | 0.680 | 0.622 | - | 0.668 |
| | | MT-late-GWMAC | 0.724 | 0.663 | **0.675** | - | 0.681 | 0.622 | - | 0.673 |
| | | MT-late-RegBN | **0.731** | 0.649 | 0.668 | - | 0.668 | 0.623 | - | 0.668 |
| | | MT-late-MIB | 0.727 | 0.665 | 0.662 | - | 0.675 | 0.640 | - | 0.674 |
| | | MT-late-WMA | 0.728 | **0.671** | 0.667 | - | **0.692** | **0.663** | - | **0.684** |
| MM-IMDb | F1↑ | MT-late | 0.602 | 0.469 | 0.488 | - | 0.553 | 0.414 | - | 0.505 |
| | | MT-late-GWMAC | 0.603 | 0.474 | 0.495 | - | **0.559** | 0.419 | - | 0.510 |
| | | MT-late-RegBN | **0.612** | 0.458 | 0.505 | - | 0.550 | 0.411 | - | 0.507 |
| | | MT-late-MIB | **0.612** | 0.453 | 0.500 | - | 0.549 | 0.415 | - | 0.506 |
| | | MT-late-WMA | 0.610 | **0.489** | **0.510** | - | **0.559** | **0.426** | - | **0.519** |
| UR-FUNNY | AUROC↑ | MT-late | 0.700 | 0.651 | 0.686 | 0.693 | 0.654 | 0.641 | 0.676 | 0.672 |
| | | MT-late-GWMAC | 0.700 | 0.653 | 0.692 | 0.699 | 0.658 | 0.646 | 0.680 | 0.675 |
| | | MT-late-RegBN | 0.709 | 0.624 | 0.694 | 0.683 | 0.645 | 0.664 | 0.671 | 0.670 |
| | | MT-late-MIB | **0.712** | 0.627 | 0.694 | 0.692 | 0.643 | 0.656 | 0.676 | 0.671 |
| | | MT-late-WMA | 0.710 | **0.660** | **0.698** | **0.702** | **0.660** | **0.672** | **0.683** | **0.684** |
| MOSEI | MAE↓ | MT-late | 0.638 | 0.665 | 0.826 | 0.642 | 0.638 | 0.822 | 0.640 | 0.696 |
| | | MT-late-GWMAC | 0.627 | 0.661 | 0.811 | 0.630 | 0.633 | 0.812 | 0.633 | 0.687 |
| | | MT-late-RegBN | 0.610 | 0.683 | 0.829 | 0.627 | 0.629 | 0.829 | 0.637 | 0.692 |
| | | MT-late-MIB | 0.610 | 0.684 | 0.844 | 0.619 | 0.624 | 0.824 | 0.626 | 0.690 |
| | | MT-late-WMA | **0.608** | **0.643** | **0.807** | **0.615** | **0.612** | **0.801** | **0.609** | **0.671** |
| Medfuse-I | AUROC↑ | MT-early | 0.848 | 0.781 | 0.742 | - | - | - | - | 0.790 |
| | | MT-early-GWMAC | 0.853 | 0.794 | 0.758 | - | - | - | - | 0.802 |
| | | MT-early-RegBN | 0.859 | 0.785 | 0.743 | - | - | - | - | 0.796 |
| | | MT-early-MIB | 0.860 | 0.774 | 0.738 | - | - | - | - | 0.791 |
| | | MT-early-WMA | **0.864** | **0.813** | **0.775** | - | - | - | - | **0.817** |

while the robustness is compromised. We attribute it to the fact that both MIB and RegBN aim to learn a multimodal representation without redundant information, making the model mainly learn from a certain modality. In our proposed WMA, the model adaptively achieves the proper modality alignment to balance multiple modalities to improve both performance and robustness.

- In summary, our WMA method simultaneously improves the performance and robustness of the multimodal transformer, outperforming all baselines across all datasets and fusion structures. In addition, compared to most parametric methods, WMA does not introduce any additional trainable parameters.

We also report all the searching results under various combinations of $\alpha$ (weight for the WMA loss) and $\beta$ (weight for the target distance) in Table 5-8 for early fusion and Table 9-12 for late fusion. We have the following observations:

- The modality alignment can be effectively adjusted by using our proposed WMA method. The obvious performance improvements can be achieved in almost half of the settings.

- The performance and robustness of the model are not exactly positively/negatively correlated with the value of $\beta$. Different tasks and even different fusion structures require different degrees of modality alignment. For example, Table 5 shows better performance and robustness at bigger target OT values for the Hateful Memes dataset under early fusion and the opposite trend is observed from Table 9 for Hateful Memes datasets under late fusion. Nevertheless, our proposed method can achieve task-dependent optimal alignment.

## 4.2 Ablation study

**Minimizing the OT distance.** We compare minimizing the OT distance (used as an agent task in many pre-train tasks Kim et al. (2021)) with our proposed WMA method. We simulate this strategy by setting $\alpha$ to 1 and $\beta$ to 0. The comparison of this method with our proposed method on the Hateful Memes dataset is shown in Fig. 4. All results are shown in Table 13 and Table 14 for early fusion and late fusion, respectively. Table 13 and Table 14 indicate that our proposed WMA outperforms this strategy in most of the test cases. The average value of all test cases in WMA exceeds that in Minimizing the OT at 2.0%, 5.3%, 1.2% and 2.3% for all datasets under early fusion. For late fusion, the improvements are 1.2%, 2.0%, 19.0% and 2.6%. We assume that different tasks require different levels of modality heterogeneity and alignment,

and over-alignment could cause a loss of modality heterogeneity which might be important to the model performance and robustness. Through only two hyperparameters, our proposed WMA method adaptively achieves optimal modal alignment, guaranteeing both performance and robustness.

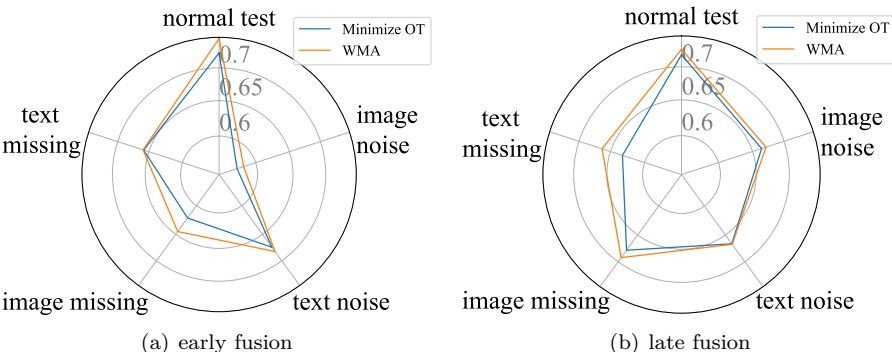

Figure 4: Results of our proposed method with minimizing OT distance on Hateful Memes datasets. See the result for all datasets in Table 13 and Table 14.

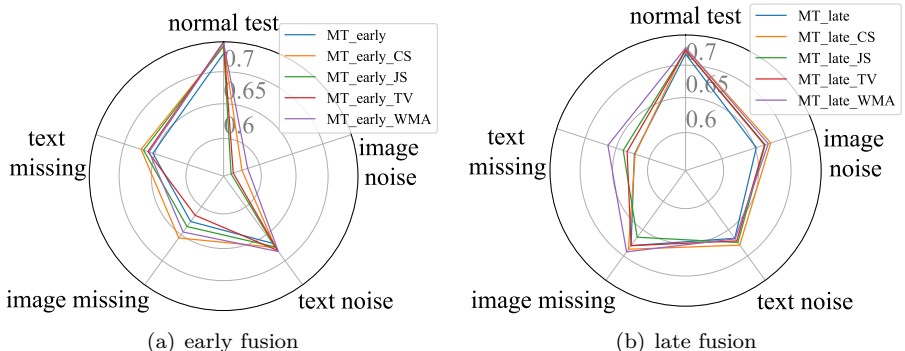

Figure 5: Results of our proposed method with different distance metrics on Hateful Memes datasets. See the result for all datasets in Table 15 and Table 16.

**Different distance metrics.** We also explore different distance metrics in the proposed regularization method. Specifically, we select the cosine similarity (CS), Jensen–Shannon (JS) divergence and total variation (TV) distance. The comparison of all methods on the Hateful Memes dataset is shown in Fig. 5. All results under early fusion and late fusion are shown in Table 15 and Table 16, respectively. From Table 15 and Table 16, we find that while the WMA method performs best across almost all datasets and fusion paradigms, some methods come close in certain datasets, i.e., cosine similarity in Hateful Memes datasets. This proves the validity of our principle idea: obtaining task-dependent optimal fusion representations by adjusting the modal alignment in the multimodal transformer.

**Computational efficiency.** To provide a more intuitive demonstration of the computational efficiency of our proposed method, we compare the training times of all methods on the UR-FUNNY and MOSEI datasets under early fusion and late fusion. Specifically, we report the time required for a training step, including both the forward and backpropagation processes for a single batch of data. As shown in Table 4, our method surpasses the strong baselines GWMAC, RegBN, and MIB in terms of training speed. This improvement can be attributed to the fact that GWMAC, RegBN, and MIB introduce additional parameters, while our WMA method operates without training any new parameters. Although the fast OT method Xie et al. (2018) used in the paper increases the computation time for distance calculations, the overall impact on runtime remains minimal. During inference, our method retains the same number of parameters and execution speed as the basic MT. Therefore, the proposed WMA method demonstrates excellent computational efficiency.

**Robustness to different random seeds.** The WMA algorithm utilizes the average OT value of the first batch of samples as the search basis. To verify the robustness of the method under different random seeds, we conduct an ablation study using the UR-FUNNY dataset with early fusion. For each experiment, we use 10 different random seeds: [0, 10, 20, 30, 40, 50, 60, 70, 80, 90]. We report the average value along with the standard deviation across all baselines in Table 3. From the table, we observe that the results of all methods exhibit slight variations under different random seeds. However, the proposed WMA method consistently demonstrates a clear advantage in terms of both performance and robustness. Additionally, we find that the optimal values for $\alpha$ and $\beta$ in the WMA method vary slightly depending on the seed (e.g., [$\alpha=5$, $\beta=1.8$] for random seed = 0, and [$\alpha=1$, $\beta=1.2$] for random seed = 20). This adaptability highlights the WMA method's strength in achieving optimal modality alignment under varying conditions.

Table 3: The performance of different methods on UR-FUNNY dataset under early fusion with different random seeds.

| Datasets | Metric | Methods | normal test | test with modality noise | | | test with missing modality | | |
|---|---|---|---|---|---|---|---|---|---|
| | | | normal | image/video noise | text/EHR noise | audio noise | image/video missing | text missing | audio missing |
| UR-FUNNY | AUROC↑ | MT-early | 0.698 ± 0.003 | 0.642 ± 0.004 | 0.677 ± 0.003 | 0.634 ± 0.004 | 0.615 ± 0.006 | 0.673 ± 0.003 | 0.691 ± 0.006 |
| | | MT-early-GWMAC | 0.701 ± 0.005 | 0.653 ± 0.004 | 0.675 ± 0.004 | 0.628 ± 0.006 | 0.610 ± 0.008 | 0.671 ± 0.004 | 0.700 ± 0.003 |
| | | MT-early-RegBN | 0.716 ± 0.004 | 0.613 ± 0.009 | 0.663 ± 0.005 | 0.617 ± 0.011 | 0.613 ± 0.003 | 0.648 ± 0.002 | 0.696 ± 0.004 |
| | | MT-early-MIB | 0.710 ± 0.003 | 0.643 ± 0.008 | 0.682 ± 0.006 | 0.632 ± 0.005 | 0.611 ± 0.003 | 0.666 ± 0.003 | 0.692 ± 0.003 |
| | | MT-early-WMA | 0.713 ± 0.003 | 0.681 ± 0.004 | 0.688 ± 0.004 | 0.646 ± 0.004 | 0.626 ± 0.003 | 0.675 ± 0.004 | 0.708 ± 0.004 |

Table 4: Computational time of a step for different methods on UR-FUNNY and MOSEI datasets under early fusion and late fusion.

| models/datasets | early fusion | | late fusion | |
|---|---|---|---|---|
| | UR-FUNNY | MOSEI | UR-FUNNY | MOSEI |
| MT-late | 2.364s | 2.288s | 1.735s | 1.506s |
| MT-late-GWMAC | 3.290s | 2.801s | 2.172s | 2.054s |
| MT-late-RegBN | 3.962s | 3.287s | 3.298s | 2.990s |
| MT-late-MIB | 3.476s | 2.947s | 2.786s | 2.409s |
| MT-late-WMA | 2.788s | 2.570s | 2.098s | 1.920s |
| MT-early-CS | 2.381s | 2.312s | 1.778s | 1.554s |
| MT-early-JS | 2.398s | 2.472s | 1.816s | 1.609s |
| MT-early-TV | 2.409s | 2.335s | 1.983s | 1.761s |

## 5  Discussion

This paper primarily investigates the application of WMA to multimodal transformers in discriminative tasks. Additionally, we explore the potential applications of WMA in generative tasks. Through a comprehensive literature survey, we observed that current generative multimodal large language models (MLLMs) continue to rely on the transformer architecture, supplemented by connectors that bridge different modality tokens—for example, the MLP used in LLAVA Liu et al. (2024) and the Q-Former used in BLIP2 Li et al. (2023). We propose that our WMA method can be seamlessly integrated with these existing approaches for the following reasons:

- Identical frameworks. Most generative MLLMs also use the transformer architecture, but involve a decoder on the output side. There is no architectural conflict for implementing the WMA in such MLLMs.

- Non-parametric approach. WMA is a non-parametric method that does not require additional training, thereby avoiding any extra overhead in the training process of MLLMs.

- Token-level implementation. WMA operates at the token level within the input to the transformer, ensuring compatibility with the connectors employed in MLLMs.

We believe that further regularization of MLLMs using WMA will enhance their performance as well as robustness, representing a significant direction for our future work.

Multimodal systems can amplify biases in the data, i.e., focusing on a specific characteristic in hiring, or credit scoring system Geirhos et al. (2020); Adewumi et al. (2024). Here We would like to discuss the impact of the proposed methodology on this issue. From Fig. 2 we observe that MT-early heavily relies on text for the decision due to the sum of the Integrated gradients values of the text token 690.63 is much higher than that of the image token 113.28. The RegBN method alleviates the problem slightly with the numbers 602.70 and 143.68. Our proposed WMA method solves this problem – the contribution of text token and image token is almost identical–811.38 and 702.77. In WMA, the image modality has a more Gaussian-like attribution value distribution compared with others, meaning that the model exploits the weak modality to learn the task. Through this case study, we demonstrate that the proposed WMA approach enables the model to allocate more equitable attention to features across different modalities, rather than concentrating on data from a single modality. Our proposed method effectively mitigates bias arising from amplified data in multimodal systems.

## 6    Conclusion

This paper addresses a pivotal challenge in multimodal transformers: the absence of a modality alignment approach during the fine-tuning phase. We introduce a Wasserstein distance-based regularization method to adjust the modality alignment degree. The proposed method does not require training more parameters and can be easily integrated into the multimodal transformer. The experimental results demonstrate significant improvements on performance and especially robustness on both 2-modalities and 3-modalities tasks, as well as for early and late fusion paradigms. In early fusion paradigm, the averaged improvement of our method compared with the best result from all baselines over all datasets are 1.4%, 1.5%, 1.4%, 0.7% and 1.4%, respectively. For the late fusion paradigm, the improvements are 1.5%, 1.8%, 1.3%, 2.3% and 1.9%. Meanwhile, our experimental results show that modality alignment needs to be task-dependent, rather than forced alignment, i.e., minimizing the OT distance between modalities, which provides valuable insights for related work. Our future work will focus on more theoretical analyses of our proposed method.

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

# A More experimental results

We report all the searching results under various combination of $\alpha$ and $\beta$ for all datasets in Table 5-8 for early fusion and 9-12 for late fusion.

Table 5: Results of different $\alpha$ and $\beta$ on Hateful Memes dataset under early fusion. Bold indicates better or equal performance than MT-early, and red font is the weight we select.

| | test case | $\beta=0$ | $\beta=0.1$ | $\beta=0.2$ | $\beta=0.4$ | $\beta=0.6$ | $\beta=0.8$ | $\beta=1$ | $\beta=1.2$ | $\beta=1.4$ | $\beta=1.6$ | $\beta=1.8$ | $\beta=2$ | $\beta=4$ | $\beta=10$ |
|---|---|---|---|---|---|---|---|---|---|---|---|---|---|---|---|
| | normal | **0.735** | 0.728 | 0.730 | **0.735** | 0.726 | **0.739** | 0.730 | 0.728 | 0.725 | 0.724 | **0.733** | 0.721 | **0.732** | 0.714 |
| | image noise | **0.573** | **0.564** | **0.560** | **0.565** | **0.557** | **0.553** | **0.565** | **0.555** | **0.570** | **0.568** | **0.564** | **0.582** | **0.565** | **0.568** |
| $\alpha=0.1$ | text noise | **0.686** | **0.682** | **0.686** | **0.686** | **0.676** | **0.683** | **0.679** | **0.683** | **0.669** | 0.665 | **0.681** | **0.668** | **0.683** | 0.661 |
| | image missing | 0.611 | 0.618 | 0.614 | 0.618 | 0.625 | **0.631** | **0.627** | 0.621 | 0.618 | 0.617 | 0.612 | 0.602 | 0.606 | **0.649** |
| | text missing | **0.671** | **0.665** | **0.666** | **0.668** | 0.657 | **0.664** | **0.664** | **0.666** | 0.655 | 0.654 | 0.657 | 0.653 | **0.659** | 0.648 |
| | normal | 0.726 | **0.731** | **0.732** | **0.731** | **0.738** | **0.735** | **0.736** | **0.732** | **0.736** | **0.740** | **0.735** | 0.728 | 0.717 | 0.721 |
| | image noise | **0.571** | **0.572** | **0.561** | **0.554** | **0.553** | **0.563** | **0.568** | **0.563** | **0.564** | **0.571** | 0.532 | **0.564** | **0.558** | **0.548** |
| $\alpha=0.2$ | text noise | **0.678** | **0.685** | **0.682** | **0.686** | **0.692** | **0.684** | **0.689** | **0.675** | **0.682** | **0.685** | **0.679** | **0.676** | **0.677** | **0.689** |
| | image missing | 0.600 | 0.606 | 0.610 | 0.617 | 0.615 | **0.635** | **0.635** | **0.643** | **0.635** | **0.642** | **0.647** | 0.615 | 0.606 | **0.634** |
| | text missing | **0.659** | **0.664** | **0.662** | **0.666** | **0.669** | **0.669** | **0.667** | **0.668** | **0.667** | **0.662** | **0.658** | **0.662** | 0.648 | **0.654** |
| | normal | 0.724 | 0.730 | 0.729 | 0.729 | **0.732** | 0.730 | 0.730 | **0.735** | **0.736** | **0.740** | **0.748** | **0.741** | 0.725 | 0.730 |
| | image noise | **0.574** | **0.571** | **0.557** | **0.567** | **0.580** | **0.563** | **0.573** | **0.556** | **0.568** | **0.552** | **0.583** | **0.556** | **0.561** | **0.562** |
| $\alpha=1$ | text noise | **0.674** | **0.686** | **0.684** | **0.682** | **0.677** | **0.684** | **0.670** | **0.681** | **0.680** | **0.677** | **0.682** | **0.676** | **0.670** | **0.675** |
| | image missing | 0.621 | 0.608 | 0.610 | 0.613 | 0.613 | 0.618 | 0.620 | 0.624 | 0.626 | 0.625 | **0.645** | 0.620 | 0.607 | 0.616 |
| | text missing | **0.658** | **0.669** | **0.666** | **0.670** | **0.666** | **0.663** | **0.662** | **0.662** | **0.660** | **0.656** | **0.659** | **0.665** | 0.648 | 0.647 |
| | normal | 0.728 | 0.721 | 0.727 | **0.733** | 0.732 | **0.734** | **0.738** | **0.733** | **0.735** | 0.721 | **0.741** | **0.737** | **0.731** | 0.728 |
| | image noise | **0.577** | **0.578** | **0.575** | **0.561** | **0.571** | **0.574** | **0.569** | 0.546 | **0.564** | **0.562** | **0.575** | **0.559** | **0.577** | 0.546 |
| $\alpha=5$ | text noise | **0.670** | 0.664 | 0.663 | **0.680** | **0.686** | **0.689** | **0.676** | **0.681** | **0.684** | **0.675** | **0.689** | **0.673** | 0.665 | **0.675** |
| | image missing | 0.621 | 0.612 | 0.621 | 0.623 | 0.623 | 0.621 | **0.632** | 0.617 | 0.620 | 0.603 | 0.624 | 0.626 | **0.646** | 0.621 |
| | text missing | **0.658** | 0.652 | **0.660** | **0.661** | **0.662** | **0.668** | **0.660** | **0.657** | **0.666** | 0.652 | **0.663** | 0.652 | **0.655** | **0.667** |
| | normal | 0.720 | 0.726 | 0.721 | 0.718 | 0.730 | 0.724 | **0.732** | 0.725 | 0.724 | **0.732** | **0.736** | **0.732** | 0.716 | 0.705 |
| | image noise | **0.576** | **0.563** | **0.580** | **0.580** | **0.557** | **0.555** | **0.555** | **0.571** | **0.555** | **0.570** | **0.576** | **0.567** | **0.577** | 0.529 |
| $\alpha=10$ | text noise | **0.672** | **0.667** | **0.668** | 0.663 | **0.676** | **0.674** | **0.681** | **0.677** | **0.671** | **0.682** | **0.669** | **0.673** | **0.669** | 0.655 |
| | image missing | **0.634** | 0.623 | 0.611 | 0.619 | **0.641** | 0.615 | 0.614 | **0.632** | **0.643** | 0.626 | 0.620 | 0.619 | **0.651** | 0.612 |
| | text missing | **0.661** | **0.660** | 0.651 | 0.647 | **0.662** | **0.658** | **0.667** | **0.655** | 0.653 | **0.660** | 0.652 | **0.659** | **0.658** | 0.636 |

Table 6: Results of different $\alpha$ and $\beta$ on MM-IMDb dataset under early fusion. Bold indicates better or equal performance than MT-early, and red font is the weight we select.

| | test case | $\beta=0$ | $\beta=0.1$ | $\beta=0.2$ | $\beta=0.4$ | $\beta=0.6$ | $\beta=0.8$ | $\beta=1$ | $\beta=1.2$ | $\beta=1.4$ | $\beta=1.6$ | $\beta=1.8$ | $\beta=2$ | $\beta=4$ | $\beta=10$ |
|---|---|---|---|---|---|---|---|---|---|---|---|---|---|---|---|
| | normal | **0.552** | 0.549 | 0.548 | 0.551 | **0.552** | 0.551 | 0.550 | 0.550 | 0.550 | **0.553** | 0.548 | 0.551 | 0.546 | 0.551 |
| | image noise | **0.269** | **0.346** | **0.317** | **0.316** | **0.304** | **0.319** | **0.298** | **0.252** | **0.284** | **0.314** | **0.273** | **0.263** | 0.243 | 0.227 |
| $\alpha=0.1$ | text noise | 0.461 | 0.456 | 0.456 | **0.464** | 0.461 | 0.458 | 0.457 | 0.456 | 0.459 | **0.464** | 0.454 | 0.458 | 0.448 | 0.458 |
| | image missing | 0.464 | 0.462 | **0.466** | 0.456 | **0.470** | 0.463 | **0.469** | **0.472** | 0.454 | **0.477** | 0.470 | **0.476** | **0.471** | 0.451 |
| | text missing | **0.422** | **0.418** | **0.415** | **0.414** | 0.412 | **0.414** | 0.408 | 0.412 | **0.414** | **0.417** | 0.410 | 0.411 | 0.412 | **0.413** |
| | normal | 0.549 | **0.552** | **0.552** | 0.550 | 0.550 | 0.551 | 0.550 | 0.551 | 0.548 | 0.551 | 0.548 | 0.550 | 0.550 | 0.548 |
| | image noise | **0.326** | **0.329** | **0.271** | **0.292** | **0.321** | 0.226 | **0.275** | **0.281** | **0.326** | **0.288** | **0.294** | **0.323** | 0.175 | 0.226 |
| $\alpha=0.2$ | text noise | 0.461 | **0.465** | 0.462 | 0.461 | 0.461 | 0.461 | 0.458 | 0.459 | 0.456 | 0.459 | 0.455 | 0.450 | 0.452 | 0.462 |
| | image missing | **0.465** | 0.457 | **0.474** | **0.468** | **0.466** | **0.466** | 0.461 | 0.460 | **0.472** | **0.474** | **0.480** | **0.484** | 0.445 | 0.452 |
| | text missing | **0.417** | **0.419** | **0.422** | 0.412 | **0.415** | 0.413 | 0.412 | **0.418** | 0.412 | **0.417** | 0.410 | 0.407 | 0.406 | **0.421** |
| | normal | 0.551 | **0.553** | **0.552** | 0.548 | **0.552** | 0.549 | 0.550 | 0.549 | 0.550 | 0.546 | 0.545 | 0.544 | 0.546 | 0.545 |
| | image noise | **0.309** | **0.276** | **0.286** | **0.278** | **0.296** | 0.237 | **0.329** | 0.247 | **0.270** | **0.269** | **0.272** | 0.235 | 0.227 | **0.291** |
| $\alpha=1$ | text noise | 0.462 | 0.459 | 0.458 | 0.458 | 0.462 | 0.459 | 0.457 | 0.457 | 0.459 | 0.449 | 0.445 | 0.449 | 0.456 | 0.452 |
| | image missing | 0.453 | 0.444 | 0.453 | 0.454 | **0.470** | 0.466 | **0.482** | 0.469 | 0.461 | **0.480** | 0.472 | 0.466 | 0.459 | 0.444 |
| | text missing | **0.415** | **0.415** | **0.415** | 0.409 | **0.421** | **0.415** | **0.416** | 0.410 | **0.415** | 0.412 | **0.413** | 0.410 | **0.414** | 0.408 |
| | normal | **0.555** | **0.558** | **0.553** | **0.553** | 0.548 | 0.550 | 0.551 | 0.549 | 0.545 | 0.544 | 0.545 | 0.545 | 0.544 | 0.544 |
| | image noise | **0.263** | **0.287** | 0.226 | **0.279** | **0.276** | **0.335** | **0.344** | **0.257** | **0.250** | **0.278** | **0.271** | 0.206 | **0.279** | 0.246 |
| $\alpha=5$ | text noise | 0.461 | **0.470** | 0.460 | 0.460 | 0.455 | 0.454 | 0.455 | 0.454 | 0.462 | 0.452 | 0.453 | 0.451 | 0.448 | 0.450 |
| | image missing | **0.470** | 0.459 | **0.468** | **0.470** | **0.472** | **0.476** | **0.488** | **0.486** | **0.472** | **0.481** | **0.475** | **0.465** | 0.441 | 0.447 |
| | text missing | **0.420** | **0.432** | **0.423** | 0.416 | 0.417 | 0.414 | **0.419** | 0.417 | 0.413 | 0.411 | 0.416 | **0.419** | 0.400 | 0.406 |
| | normal | **0.555** | **0.563** | **0.553** | **0.554** | **0.554** | 0.548 | **0.552** | 0.551 | 0.548 | 0.544 | 0.548 | 0.550 | 0.549 | 0.540 |
| | image noise | **0.282** | **0.355** | **0.254** | **0.332** | 0.243 | 0.239 | 0.238 | 0.250 | 0.219 | **0.298** | 0.206 | 0.222 | **0.284** | 0.215 |
| $\alpha=10$ | text noise | **0.469** | **0.471** | **0.463** | 0.461 | **0.464** | 0.457 | 0.456 | 0.462 | 0.461 | 0.453 | 0.451 | 0.453 | 0.456 | 0.433 |
| | image missing | **0.473** | **0.477** | **0.481** | **0.473** | 0.469 | 0.469 | **0.474** | 0.470 | 0.470 | 0.476 | 0.473 | **0.479** | 0.469 | 0.435 |
| | text missing | **0.430** | **0.437** | **0.429** | 0.421 | 0.420 | 0.405 | 0.415 | 0.416 | 0.425 | 0.413 | 0.427 | 0.425 | 0.415 | 0.396 |

Table 7: Results of different $\alpha$ and $\beta$ on UR-FUNNY dataset under early fusion. Bold indicates better or equal performance than MT-early, and red font is the weight we select.

| | test case | $\beta=0$ | $\beta=0.1$ | $\beta=0.2$ | $\beta=0.4$ | $\beta=0.6$ | $\beta=0.8$ | $\beta=1$ | $\beta=1.2$ | $\beta=1.4$ | $\beta=1.6$ | $\beta=1.8$ | $\beta=2$ | $\beta=4$ | $\beta=10$ |
|---|---|---|---|---|---|---|---|---|---|---|---|---|---|---|---|
| | normal | **0.702** | **0.701** | **0.701** | 0.700 | 0.698 | 0.700 | **0.702** | **0.704** | **0.704** | 0.699 | 0.699 | 0.699 | 0.698 | 0.697 |
| | image noise | **0.646** | **0.648** | 0.643 | **0.649** | **0.647** | **0.654** | **0.652** | **0.653** | **0.652** | **0.651** | **0.648** | **0.654** | **0.656** | **0.657** |
| $\alpha=0.1$ | text noise | **0.680** | **0.682** | **0.681** | **0.681** | **0.680** | **0.679** | **0.684** | **0.683** | **0.686** | **0.681** | **0.680** | 0.674 | **0.683** | 0.670 |
| | audio noise | 0.630 | 0.633 | 0.630 | 0.635 | 0.630 | 0.633 | 0.626 | 0.631 | 0.633 | 0.629 | 0.627 | 0.635 | **0.643** | 0.634 |
| | image missing | 0.608 | 0.608 | 0.609 | 0.606 | 0.609 | 0.608 | **0.613** | **0.614** | **0.614** | 0.608 | 0.612 | 0.597 | 0.611 | 0.601 |
| | text missing | **0.673** | **0.682** | **0.684** | 0.669 | **0.674** | 0.668 | **0.682** | **0.678** | **0.680** | 0.670 | **0.682** | 0.647 | 0.666 | 0.668 |
| | audio missing | **0.700** | **0.699** | **0.697** | 0.696 | 0.694 | **0.697** | **0.697** | **0.699** | **0.701** | 0.696 | 0.695 | 0.691 | 0.696 | 0.692 |
| | normal | 0.699 | **0.701** | 0.699 | **0.701** | 0.698 | 0.699 | **0.704** | **0.704** | **0.702** | **0.702** | 0.699 | **0.702** | **0.705** | 0.700 |
| | image noise | 0.644 | **0.648** | **0.653** | **0.653** | **0.653** | **0.648** | 0.645 | **0.650** | **0.651** | **0.655** | **0.658** | 0.643 | **0.646** | **0.662** |
| $\alpha=0.2$ | text noise | **0.680** | **0.682** | 0.678 | **0.683** | **0.683** | **0.684** | **0.681** | **0.683** | **0.680** | 0.678 | **0.682** | **0.681** | 0.672 | **0.681** |
| | audio noise | 0.626 | 0.630 | 0.632 | **0.636** | **0.638** | 0.631 | 0.626 | 0.629 | 0.627 | 0.632 | 0.632 | 0.620 | 0.620 | 0.635 |
| | image missing | 0.603 | 0.606 | 0.604 | 0.611 | 0.609 | 0.610 | 0.610 | 0.612 | 0.610 | 0.609 | 0.606 | 0.610 | 0.604 | **0.627** |
| | text missing | **0.682** | **0.685** | 0.677 | **0.680** | 0.670 | **0.677** | **0.678** | 0.672 | **0.683** | 0.672 | 0.669 | **0.675** | 0.666 | **0.689** |
| | audio missing | 0.695 | **0.698** | 0.695 | **0.697** | 0.695 | 0.695 | **0.699** | **0.699** | **0.697** | **0.699** | 0.696 | **0.698** | **0.700** | 0.696 |
| | normal | **0.705** | 0.700 | **0.702** | 0.699 | 0.699 | **0.701** | **0.701** | 0.695 | **0.702** | **0.705** | **0.702** | **0.705** | **0.704** | **0.704** |
| | image noise | **0.661** | **0.665** | **0.653** | 0.648 | **0.656** | 0.651 | 0.649 | 0.644 | 0.638 | **0.660** | 0.649 | **0.657** | **0.666** | **0.671** |
| $\alpha=1$ | text noise | **0.679** | **0.683** | **0.679** | **0.681** | **0.679** | **0.683** | **0.679** | 0.676 | **0.680** | 0.667 | **0.686** | **0.688** | 0.667 | **0.686** |
| | audio noise | 0.633 | **0.643** | 0.634 | 0.635 | 0.627 | 0.634 | 0.631 | 0.629 | 0.634 | **0.636** | 0.630 | **0.639** | **0.640** | 0.633 |
| | image missing | **0.628** | 0.608 | 0.604 | 0.607 | 0.608 | 0.607 | **0.613** | 0.602 | 0.609 | 0.593 | 0.611 | **0.616** | **0.621** | **0.620** |
| | text missing | 0.658 | 0.664 | **0.675** | 0.667 | **0.681** | **0.682** | 0.653 | **0.676** | **0.675** | 0.582 | **0.671** | **0.680** | 0.631 | 0.647 |
| | audio missing | **0.699** | **0.697** | **0.698** | 0.695 | 0.696 | **0.698** | **0.697** | 0.693 | **0.699** | **0.704** | **0.698** | **0.702** | **0.697** | **0.700** |
| | normal | **0.704** | 0.695 | 0.699 | 0.697 | 0.637 | **0.706** | **0.702** | **0.705** | **0.701** | **0.702** | **0.712** | 0.699 | **0.704** | 0.617 |
| | image noise | 0.644 | **0.671** | **0.667** | **0.662** | 0.605 | 0.645 | 0.638 | **0.655** | **0.652** | **0.657** | **0.681** | **0.671** | **0.671** | 0.614 |
| $\alpha=5$ | text noise | 0.675 | 0.661 | 0.676 | **0.679** | 0.624 | 0.656 | **0.679** | **0.682** | **0.685** | **0.688** | **0.686** | **0.684** | **0.681** | 0.621 |
| | audio noise | 0.631 | 0.635 | **0.641** | **0.644** | 0.616 | 0.623 | 0.624 | **0.637** | 0.633 | 0.632 | **0.643** | **0.637** | 0.634 | 0.598 |
| | image missing | 0.604 | 0.606 | 0.611 | 0.608 | 0.555 | **0.619** | 0.604 | 0.609 | **0.620** | **0.623** | **0.624** | 0.605 | **0.620** | 0.549 |
| | text missing | 0.659 | 0.644 | 0.667 | 0.669 | 0.615 | 0.600 | **0.675** | **0.673** | 0.661 | **0.672** | **0.674** | **0.672** | 0.643 | 0.555 |
| | audio missing | **0.703** | 0.693 | 0.696 | 0.695 | 0.618 | **0.702** | **0.697** | **0.702** | **0.698** | **0.698** | **0.703** | 0.696 | **0.700** | 0.596 |
| | normal | 0.698 | **0.704** | 0.698 | 0.699 | **0.701** | 0.700 | 0.696 | **0.703** | 0.699 | **0.701** | **0.704** | 0.696 | 0.698 | **0.709** |
| | image noise | **0.665** | **0.675** | **0.667** | 0.653 | **0.669** | 0.659 | 0.642 | **0.662** | **0.668** | **0.671** | 0.666 | **0.671** | **0.668** | 0.663 |
| $\alpha=10$ | text noise | 0.676 | **0.681** | **0.685** | 0.666 | 0.675 | **0.679** | **0.680** | **0.683** | **0.683** | **0.690** | **0.683** | **0.682** | 0.674 | 0.664 |
| | audio noise | **0.637** | **0.649** | **0.646** | 0.631 | **0.645** | **0.647** | 0.623 | 0.635 | 0.635 | **0.638** | **0.637** | 0.634 | **0.661** | 0.620 |
| | image missing | 0.605 | **0.629** | **0.613** | 0.611 | 0.604 | **0.618** | 0.600 | **0.643** | **0.621** | 0.619 | 0.612 | **0.626** | **0.639** | 0.602 |
| | text missing | 0.659 | 0.667 | **0.673** | 0.652 | 0.657 | 0.665 | **0.675** | 0.662 | 0.658 | **0.692** | 0.651 | 0.662 | 0.626 | 0.646 |
| | audio missing | 0.695 | **0.699** | 0.696 | 0.695 | **0.699** | **0.697** | 0.693 | **0.699** | 0.696 | 0.696 | **0.700** | 0.692 | 0.687 | **0.706** |

Table 8: Results of different $\alpha$ and $\beta$ on MOSEI dataset under early fusion. Bold indicates better or equal performance than MT-early, and red font is the weight we select.

| | test case | $\beta=0$ | $\beta=0.1$ | $\beta=0.2$ | $\beta=0.4$ | $\beta=0.6$ | $\beta=0.8$ | $\beta=1$ | $\beta=1.2$ | $\beta=1.4$ | $\beta=1.6$ | $\beta=1.8$ | $\beta=2$ | $\beta=4$ | $\beta=10$ |
|---|---|---|---|---|---|---|---|---|---|---|---|---|---|---|---|
| | normal | 0.668 | **0.628** | 0.647 | **0.627** | **0.624** | **0.626** | 0.639 | 0.647 | **0.622** | 0.654 | 0.635 | 0.634 | 0.635 | 0.769 |
| | image noise | 0.725 | 0.697 | **0.688** | 0.700 | **0.678** | 0.725 | 0.709 | 0.731 | 0.741 | 0.708 | 0.724 | 0.739 | 0.737 | 0.784 |
| $\alpha=0.1$ | text noise | 0.849 | 0.863 | 0.844 | **0.813** | **0.828** | **0.832** | **0.833** | 0.839 | **0.822** | 0.857 | **0.826** | **0.825** | **0.827** | 0.903 |
| | audio noise | 0.691 | 0.678 | 0.674 | **0.665** | **0.653** | 0.689 | 0.696 | 0.693 | 0.678 | 0.685 | 0.679 | 0.692 | 0.691 | 0.769 |
| | image missing | 0.678 | **0.635** | 0.653 | **0.642** | **0.629** | **0.635** | **0.648** | 0.652 | **0.632** | 0.655 | **0.646** | **0.646** | **0.644** | 0.771 |
| | text missing | 0.829 | 0.835 | 0.828 | **0.813** | **0.818** | **0.825** | **0.821** | **0.825** | **0.819** | 0.838 | **0.821** | **0.818** | 0.827 | 0.872 |
| | audio missing | 0.669 | **0.630** | 0.651 | **0.628** | **0.628** | **0.623** | 0.640 | 0.649 | **0.624** | 0.653 | 0.638 | **0.633** | 0.650 | 0.792 |
| | normal | **0.624** | **0.628** | 0.634 | 0.643 | **0.630** | **0.626** | **0.622** | **0.628** | 0.657 | 0.636 | **0.623** | **0.629** | 0.652 | 0.635 |
| | image noise | 0.722 | **0.686** | 0.691 | 0.692 | 0.699 | 0.735 | 0.703 | **0.680** | 0.737 | 0.741 | 0.720 | 0.709 | 0.721 | 0.745 |
| $\alpha=0.2$ | text noise | 0.834 | 0.834 | **0.830** | 0.852 | **0.824** | **0.823** | 0.847 | 0.850 | 0.912 | **0.830** | **0.816** | **0.811** | 0.847 | **0.830** |
| | audio noise | 0.686 | **0.654** | 0.681 | 0.674 | 0.679 | 0.687 | 0.691 | 0.695 | 0.731 | 0.692 | **0.657** | **0.668** | 0.687 | 0.695 |
| | image missing | **0.634** | **0.635** | **0.637** | 0.653 | **0.632** | **0.635** | **0.633** | **0.633** | 0.657 | **0.645** | **0.636** | **0.641** | 0.656 | **0.644** |
| | text missing | **0.822** | **0.821** | **0.817** | 0.830 | **0.820** | **0.817** | 0.832 | 0.827 | 0.868 | **0.821** | **0.820** | **0.821** | 0.827 | **0.822** |
| | audio missing | **0.628** | 0.649 | 0.640 | 0.650 | **0.629** | **0.627** | **0.628** | 0.636 | 0.656 | 0.641 | **0.634** | 0.638 | 0.668 | 0.652 |
| | normal | **0.624** | 0.689 | **0.628** | **0.615** | **0.630** | 0.833 | **0.632** | 0.640 | **0.621** | 0.642 | 0.642 | 0.650 | 0.653 | 0.829 |
| | image noise | 0.711 | 0.740 | 0.732 | 0.721 | 0.698 | 0.838 | 0.724 | 0.707 | 0.710 | 0.733 | 0.715 | 0.755 | **0.688** | 0.834 |
| $\alpha=1$ | text noise | 0.855 | **0.833** | 0.847 | **0.827** | 0.845 | 0.836 | **0.827** | 0.838 | **0.821** | **0.822** | **0.821** | 0.869 | 0.843 | 0.836 |
| | audio noise | 0.676 | 0.710 | 0.684 | **0.662** | **0.655** | 0.839 | 0.679 | **0.664** | **0.671** | **0.664** | 0.679 | 0.712 | 0.676 | 0.834 |
| | image missing | **0.629** | 0.694 | **0.635** | **0.621** | **0.640** | 0.833 | **0.640** | **0.647** | **0.627** | 0.655 | **0.648** | 0.658 | 0.656 | 0.830 |
| | text missing | 0.831 | **0.819** | 0.832 | **0.817** | 0.826 | 0.835 | **0.817** | **0.824** | **0.820** | **0.819** | **0.816** | 0.841 | 0.876 | 0.835 |
| | audio missing | **0.623** | 0.690 | **0.627** | **0.621** | 0.637 | 0.837 | **0.634** | 0.646 | **0.634** | 0.646 | 0.658 | 0.649 | 0.718 | 0.990 |
| | normal | **0.632** | 0.658 | **0.615** | **0.631** | **0.627** | **0.624** | 0.643 | **0.624** | **0.627** | 0.658 | 0.640 | 0.642 | 0.633 | 0.832 |
| | image noise | 0.743 | 0.710 | **0.676** | 0.710 | 0.691 | **0.687** | 0.697 | 0.714 | 0.707 | 0.716 | 0.743 | 0.736 | **0.686** | 0.834 |
| $\alpha=5$ | text noise | **0.818** | 0.865 | **0.824** | 0.835 | 0.849 | 0.841 | 0.839 | 0.837 | 0.852 | 0.860 | 0.871 | 0.855 | **0.823** | 0.834 |
| | audio noise | 0.685 | 0.685 | **0.664** | 0.676 | **0.669** | **0.666** | **0.663** | 0.675 | 0.678 | 0.680 | 0.689 | 0.695 | **0.664** | 0.834 |
| | image missing | **0.643** | 0.660 | **0.619** | **0.636** | **0.635** | **0.635** | 0.653 | **0.631** | **0.637** | 0.667 | 0.652 | 0.652 | **0.643** | 0.833 |
| | text missing | **0.812** | 0.842 | **0.813** | **0.821** | 0.829 | **0.824** | 0.826 | **0.824** | 0.830 | 0.836 | 0.844 | 0.871 | 0.910 | 0.834 |
| | audio missing | **0.630** | 0.654 | **0.623** | 0.639 | 0.641 | **0.629** | 0.646 | **0.633** | **0.628** | 0.656 | 0.641 | 0.643 | 0.897 | 0.913 |
| | normal | **0.629** | **0.626** | 0.655 | **0.619** | **0.627** | **0.627** | 0.634 | **0.631** | **0.622** | 0.650 | **0.632** | **0.621** | 0.648 | 0.832 |
| | image noise | 0.707 | 0.717 | 0.718 | 0.728 | 0.740 | 0.735 | 0.700 | 0.720 | 0.698 | 0.715 | 0.717 | 0.703 | 0.705 | 0.833 |
| $\alpha=10$ | text noise | 0.834 | **0.815** | 0.860 | 0.825 | 0.816 | **0.824** | 0.840 | **0.820** | 0.829 | 0.911 | 0.834 | **0.831** | **0.823** | **0.833** |
| | audio noise | **0.665** | **0.661** | 0.674 | **0.667** | 0.696 | 0.692 | 0.681 | 0.674 | 0.680 | 0.674 | **0.699** | **0.667** | 0.682 | 0.833 |
| | image missing | **0.632** | **0.629** | 0.661 | **0.626** | **0.631** | **0.643** | **0.634** | **0.636** | **0.626** | 0.657 | **0.639** | **0.625** | 0.665 | 0.832 |
| | text missing | **0.822** | **0.809** | 0.838 | **0.814** | **0.812** | **0.817** | 0.828 | **0.816** | **0.820** | 0.869 | 0.829 | 0.838 | 0.904 | 0.832 |
| | audio missing | **0.628** | **0.626** | 0.665 | **0.628** | **0.626** | 0.641 | 0.647 | 0.637 | **0.631** | 0.650 | **0.634** | **0.624** | 1.029 | 0.834 |

Table 9: Results of different $\alpha$ and $\beta$ on Hateful Memes dataset under late fusion. Bold indicates better or equal performance than MT-early, and red font is the weight we select.

| | test case | $\beta=0$ | $\beta=0.1$ | $\beta=0.2$ | $\beta=0.4$ | $\beta=0.6$ | $\beta=0.8$ | $\beta=1$ | $\beta=1.2$ | $\beta=1.4$ | $\beta=1.6$ | $\beta=1.8$ | $\beta=2$ | $\beta=4$ | $\beta=10$ |
|---|---|---|---|---|---|---|---|---|---|---|---|---|---|---|---|
| | normal | 0.685 | **0.724** | **0.725** | **0.722** | **0.722** | **0.723** | 0.680 | 0.680 | 0.681 | 0.710 | 0.707 | **0.721** | 0.709 | 0.686 |
| | image noise | **0.661** | **0.666** | **0.666** | 0.663 | **0.662** | **0.662** | 0.658 | 0.658 | 0.659 | **0.661** | 0.659 | **0.672** | 0.654 | **0.665** |
| $\alpha=0.1$ | text noise | 0.636 | **0.670** | **0.672** | **0.670** | **0.671** | **0.671** | 0.633 | 0.632 | 0.633 | 0.654 | 0.663 | **0.668** | 0.652 | 0.635 |
| | image missing | 0.674 | 0.680 | **0.682** | 0.680 | 0.674 | **0.687** | 0.671 | 0.672 | 0.672 | **0.687** | **0.685** | **0.692** | 0.678 | 0.673 |
| | text missing | 0.583 | **0.636** | **0.636** | **0.633** | **0.632** | 0.610 | 0.581 | 0.579 | 0.580 | 0.594 | 0.594 | 0.614 | 0.606 | 0.576 |
| | normal | 0.685 | 0.684 | 0.684 | 0.685 | **0.724** | **0.723** | 0.680 | 0.681 | 0.708 | **0.722** | **0.723** | **0.721** | 0.707 | 0.715 |
| | image noise | 0.647 | **0.660** | **0.660** | 0.661 | **0.665** | 0.656 | 0.658 | 0.659 | 0.660 | **0.672** | **0.674** | **0.675** | **0.664** | **0.674** |
| $\alpha=0.2$ | text noise | 0.650 | 0.635 | 0.636 | 0.636 | **0.672** | **0.671** | 0.633 | 0.633 | 0.658 | **0.670** | **0.669** | 0.666 | 0.658 | 0.661 |
| | image missing | 0.648 | 0.673 | 0.673 | 0.674 | **0.684** | **0.683** | 0.671 | 0.671 | 0.680 | **0.697** | **0.696** | **0.694** | **0.681** | **0.683** |
| | text missing | 0.607 | 0.583 | 0.583 | 0.583 | 0.629 | 0.630 | 0.580 | 0.581 | 0.580 | 0.606 | 0.618 | 0.609 | 0.622 | 0.609 |
| | normal | **0.719** | 0.712 | 0.717 | **0.722** | 0.684 | 0.680 | **0.726** | **0.723** | 0.716 | 0.718 | 0.688 | **0.722** | 0.589 | 0.594 |
| | image noise | **0.664** | 0.652 | **0.657** | **0.665** | 0.661 | 0.659 | **0.671** | **0.672** | 0.669 | 0.663 | **0.664** | **0.675** | 0.569 | 0.571 |
| $\alpha=1$ | text noise | 0.666 | 0.662 | **0.667** | **0.673** | 0.635 | 0.633 | **0.675** | **0.670** | 0.662 | **0.670** | 0.637 | 0.664 | 0.567 | 0.574 |
| | image missing | 0.678 | 0.675 | 0.673 | 0.680 | 0.674 | 0.672 | **0.697** | **0.694** | **0.693** | **0.684** | 0.668 | **0.684** | 0.577 | 0.577 |
| | text missing | **0.632** | 0.609 | **0.625** | **0.623** | 0.582 | 0.580 | 0.613 | **0.623** | 0.604 | 0.616 | 0.584 | **0.624** | 0.562 | 0.567 |
| | normal | 0.716 | 0.717 | 0.717 | 0.713 | 0.717 | 0.712 | 0.709 | **0.722** | **0.723** | **0.719** | 0.676 | 0.589 | 0.599 | 0.601 |
| | image noise | **0.659** | **0.671** | **0.674** | 0.661 | **0.659** | 0.664 | **0.660** | **0.676** | 0.666 | 0.656 | **0.659** | 0.570 | 0.569 | 0.560 |
| $\alpha=5$ | text noise | 0.658 | 0.659 | 0.660 | 0.658 | 0.663 | **0.669** | 0.661 | 0.666 | 0.664 | 0.664 | 0.626 | 0.567 | 0.580 | 0.584 |
| | image missing | 0.678 | **0.691** | **0.695** | 0.680 | **0.684** | **0.687** | **0.685** | **0.695** | 0.678 | **0.685** | 0.667 | 0.577 | 0.570 | 0.549 |
| | text missing | **0.626** | **0.626** | **0.631** | 0.614 | 0.617 | 0.599 | 0.603 | 0.615 | 0.597 | **0.625** | 0.583 | 0.562 | 0.575 | 0.581 |
| | normal | **0.724** | **0.728** | **0.720** | 0.717 | 0.681 | 0.681 | 0.711 | **0.721** | 0.715 | 0.713 | 0.590 | 0.591 | 0.601 | 0.597 |
| | image noise | **0.662** | **0.671** | **0.660** | **0.671** | **0.660** | **0.661** | **0.660** | **0.676** | 0.657 | 0.654 | 0.570 | 0.570 | 0.564 | 0.551 |
| $\alpha=10$ | text noise | 0.663 | **0.667** | 0.658 | 0.660 | 0.632 | 0.634 | 0.664 | 0.665 | 0.662 | 0.665 | 0.568 | 0.570 | 0.583 | 0.582 |
| | image missing | **0.708** | **0.692** | **0.682** | **0.694** | 0.674 | 0.671 | 0.680 | **0.692** | 0.680 | 0.675 | 0.577 | 0.578 | 0.556 | 0.537 |
| | text missing | **0.646** | **0.663** | 0.632 | 0.626 | 0.584 | 0.583 | 0.606 | 0.620 | **0.626** | **0.625** | 0.564 | 0.565 | 0.580 | 0.578 |

Table 10: Results of different $\alpha$ and $\beta$ on MM-IMDb dataset under late fusion. Bold indicates better or equal performance than MT-early, and red font is the weight we select.

| | test case | $\beta=0$ | $\beta=0.1$ | $\beta=0.2$ | $\beta=0.4$ | $\beta=0.6$ | $\beta=0.8$ | $\beta=1$ | $\beta=1.2$ | $\beta=1.4$ | $\beta=1.6$ | $\beta=1.8$ | $\beta=2$ | $\beta=4$ | $\beta=10$ |
|---|---|---|---|---|---|---|---|---|---|---|---|---|---|---|---|
| | normal | **0.603** | **0.603** | 0.602 | **0.603** | 0.602 | 0.602 | 0.602 | 0.601 | 0.601 | 0.601 | 0.601 | 0.601 | 0.602 | 0.600 |
| | image noise | **0.470** | **0.470** | **0.470** | **0.471** | **0.473** | **0.474** | **0.474** | **0.473** | **0.471** | 0.469 | 0.468 | 0.466 | 0.450 | 0.427 |
| $\alpha=0.1$ | text noise | **0.498** | **0.496** | **0.495** | **0.496** | **0.495** | **0.495** | 0.494 | 0.494 | **0.494** | **0.495** | **0.495** | **0.495** | **0.497** | 0.493 |
| | image missing | **0.557** | **0.557** | **0.557** | **0.557** | **0.558** | **0.559** | **0.557** | 0.554 | 0.554 | 0.554 | 0.554 | 0.554 | **0.556** | 0.553 |
| | text missing | **0.415** | **0.416** | **0.417** | **0.416** | **0.418** | **0.419** | **0.419** | **0.417** | **0.417** | **0.417** | **0.417** | **0.417** | **0.417** | 0.411 |
| | normal | **0.603** | **0.603** | **0.603** | **0.603** | 0.600 | 0.602 | 0.602 | 0.601 | 0.602 | 0.600 | 0.600 | 0.601 | 0.601 | 0.599 |
| | image noise | **0.472** | **0.470** | **0.472** | **0.471** | **0.471** | **0.474** | **0.475** | **0.472** | 0.469 | 0.467 | 0.464 | 0.462 | 0.438 | 0.428 |
| $\alpha=0.2$ | text noise | **0.498** | **0.497** | **0.498** | **0.495** | 0.492 | **0.494** | 0.494 | **0.495** | **0.495** | **0.496** | **0.496** | **0.496** | **0.496** | 0.486 |
| | image missing | **0.555** | **0.555** | **0.556** | **0.557** | **0.555** | **0.558** | **0.558** | 0.554 | 0.554 | 0.553 | 0.553 | 0.553 | **0.560** | **0.556** |
| | text missing | **0.416** | 0.413 | **0.416** | **0.417** | **0.416** | **0.419** | **0.419** | **0.417** | **0.418** | **0.416** | **0.417** | **0.417** | **0.417** | **0.420** |
| | normal | **0.609** | **0.606** | **0.605** | **0.603** | 0.601 | 0.602 | 0.602 | 0.600 | 0.602 | **0.603** | 0.602 | 0.602 | 0.601 | 0.598 |
| | image noise | 0.463 | **0.470** | **0.472** | **0.472** | **0.472** | **0.473** | **0.474** | 0.469 | 0.464 | 0.465 | 0.466 | 0.461 | 0.441 | 0.422 |
| $\alpha=1$ | text noise | **0.506** | **0.505** | **0.502** | **0.499** | **0.494** | **0.494** | **0.496** | **0.495** | **0.497** | **0.498** | **0.497** | **0.496** | 0.486 | 0.477 |
| | image missing | **0.554** | **0.554** | **0.554** | **0.556** | **0.554** | **0.557** | **0.554** | 0.552 | 0.552 | **0.554** | **0.554** | 0.553 | 0.549 | 0.531 |
| | text missing | **0.415** | **0.417** | **0.418** | **0.417** | **0.416** | **0.418** | **0.417** | **0.417** | **0.418** | **0.419** | **0.418** | **0.416** | **0.423** | **0.423** |
| | normal | 0.602 | **0.610** | **0.609** | **0.605** | 0.600 | 0.601 | 0.601 | 0.599 | 0.600 | 0.599 | 0.599 | 0.595 | 0.592 | 0.591 |
| | image noise | 0.411 | **0.489** | 0.463 | **0.474** | **0.474** | **0.475** | **0.473** | 0.465 | 0.461 | 0.463 | 0.455 | 0.442 | 0.414 | 0.400 |
| $\alpha=5$ | text noise | 0.482 | **0.510** | **0.509** | **0.502** | **0.496** | **0.493** | **0.495** | **0.496** | **0.497** | **0.496** | **0.494** | **0.493** | 0.475 | 0.464 |
| | image missing | **0.555** | **0.559** | **0.556** | 0.553 | 0.552 | **0.555** | **0.554** | 0.551 | 0.550 | 0.551 | 0.550 | 0.542 | 0.515 | 0.507 |
| | text missing | 0.397 | **0.426** | **0.419** | **0.420** | **0.416** | **0.416** | **0.416** | **0.416** | **0.415** | 0.413 | 0.412 | 0.404 | 0.411 | 0.413 |
| | normal | 0.600 | **0.609** | **0.612** | **0.604** | 0.599 | 0.600 | 0.600 | 0.599 | 0.598 | 0.598 | 0.594 | 0.594 | 0.592 | 0.589 |
| | image noise | 0.449 | 0.432 | 0.459 | **0.474** | **0.473** | **0.475** | **0.471** | 0.464 | 0.461 | 0.458 | 0.446 | 0.435 | 0.406 | 0.405 |
| $\alpha=10$ | text noise | 0.488 | **0.501** | **0.508** | **0.502** | **0.497** | **0.494** | **0.495** | **0.496** | **0.496** | **0.492** | **0.491** | 0.485 | 0.468 | 0.460 |
| | image missing | 0.553 | **0.556** | **0.555** | 0.552 | 0.551 | **0.554** | 0.553 | 0.551 | 0.549 | 0.549 | 0.538 | 0.540 | 0.510 | 0.507 |
| | text missing | 0.393 | 0.414 | **0.422** | **0.420** | **0.416** | 0.414 | **0.416** | **0.415** | 0.413 | 0.411 | 0.401 | 0.408 | 0.413 | 0.410 |

Table 11: Results of different $\alpha$ and $\beta$ on UR-FUNNY dataset under late fusion. Bold indicates better or equal performance than MT-early, and red font is the weight we select.

| | test case | $\beta=0$ | $\beta=0.1$ | $\beta=0.2$ | $\beta=0.4$ | $\beta=0.6$ | $\beta=0.8$ | $\beta=1$ | $\beta=1.2$ | $\beta=1.4$ | $\beta=1.6$ | $\beta=1.8$ | $\beta=2$ | $\beta=4$ | $\beta=10$ |
|---|---|---|---|---|---|---|---|---|---|---|---|---|---|---|---|
| | normal | 0.631 | 0.623 | 0.590 | **0.701** | **0.701** | **0.701** | **0.701** | **0.701** | **0.702** | **0.702** | 0.700 | 0.699 | **0.702** | 0.696 |
| | image noise | 0.506 | 0.506 | 0.497 | **0.671** | **0.663** | **0.655** | **0.652** | 0.644 | 0.632 | 0.619 | 0.611 | 0.607 | 0.578 | 0.578 |
| $\alpha=0.1$ | text noise | 0.635 | 0.626 | 0.588 | **0.695** | **0.695** | **0.693** | **0.692** | **0.691** | **0.689** | 0.686 | 0.686 | **0.690** | 0.686 | 0.682 |
| | audio noise | 0.629 | 0.622 | 0.586 | **0.700** | **0.700** | **0.699** | **0.699** | **0.700** | **0.700** | **0.699** | **0.697** | **0.694** | 0.693 | 0.689 |
| | image missing | 0.551 | 0.548 | 0.540 | **0.655** | **0.656** | **0.657** | **0.658** | **0.658** | 0.654 | 0.650 | 0.651 | 0.644 | 0.646 | 0.637 |
| | text missing | 0.614 | 0.606 | 0.575 | **0.652** | **0.653** | 0.646 | **0.647** | **0.649** | **0.651** | **0.649** | **0.647** | **0.657** | **0.672** | **0.668** |
| | audio missing | 0.601 | 0.595 | 0.579 | **0.684** | **0.683** | **0.680** | **0.680** | **0.680** | **0.678** | 0.676 | **0.677** | 0.674 | 0.659 | 0.654 |
| | normal | 0.592 | 0.618 | 0.634 | 0.700 | **0.710** | **0.701** | **0.701** | **0.701** | **0.702** | 0.700 | **0.701** | **0.702** | 0.696 | 0.630 |
| | image noise | 0.495 | 0.504 | 0.505 | **0.682** | **0.660** | **0.654** | 0.651 | 0.644 | 0.633 | 0.620 | 0.606 | 0.603 | 0.579 | 0.513 |
| $\alpha=0.2$ | text noise | 0.592 | 0.620 | 0.637 | **0.696** | **0.698** | **0.693** | **0.692** | **0.691** | **0.688** | 0.686 | **0.692** | **0.690** | 0.682 | 0.634 |
| | audio noise | 0.585 | 0.614 | 0.630 | **0.701** | **0.702** | **0.699** | **0.700** | **0.699** | **0.699** | **0.698** | **0.696** | **0.694** | 0.689 | 0.631 |
| | image missing | 0.541 | 0.551 | 0.554 | 0.648 | **0.660** | **0.657** | **0.657** | **0.659** | **0.656** | 0.647 | 0.644 | 0.644 | 0.638 | 0.549 |
| | text missing | 0.580 | 0.612 | 0.625 | **0.658** | **0.672** | 0.647 | **0.648** | **0.649** | **0.653** | **0.647** | **0.661** | **0.660** | **0.667** | 0.624 |
| | audio missing | 0.586 | 0.607 | 0.611 | **0.684** | **0.683** | **0.680** | **0.680** | **0.679** | **0.677** | 0.676 | 0.675 | 0.673 | 0.652 | 0.606 |
| | normal | 0.604 | 0.617 | 0.628 | 0.616 | **0.701** | **0.703** | 0.700 | **0.704** | 0.700 | **0.702** | **0.706** | **0.702** | 0.603 | 0.596 |
| | image noise | 0.492 | 0.492 | 0.492 | 0.493 | **0.660** | **0.655** | 0.651 | 0.644 | 0.632 | 0.619 | 0.604 | 0.590 | 0.510 | 0.508 |
| $\alpha=1$ | text noise | 0.611 | 0.624 | 0.637 | 0.621 | **0.694** | **0.693** | **0.695** | **0.690** | **0.687** | **0.693** | **0.693** | **0.687** | 0.603 | 0.596 |
| | audio noise | 0.595 | 0.607 | 0.615 | 0.607 | **0.701** | **0.700** | **0.700** | **0.701** | **0.697** | **0.698** | **0.700** | **0.696** | 0.600 | 0.590 |
| | image missing | 0.542 | 0.546 | 0.549 | 0.547 | **0.655** | **0.655** | **0.656** | **0.655** | 0.643 | 0.641 | 0.640 | 0.643 | 0.544 | 0.542 |
| | text missing | 0.588 | 0.597 | 0.606 | 0.604 | **0.648** | **0.648** | **0.649** | **0.654** | **0.651** | **0.661** | **0.666** | **0.671** | 0.597 | 0.588 |
| | audio missing | 0.594 | 0.604 | 0.614 | 0.615 | **0.680** | **0.678** | **0.681** | **0.677** | **0.679** | 0.675 | 0.666 | 0.656 | 0.605 | 0.597 |
| | normal | 0.613 | 0.602 | 0.595 | 0.699 | 0.701 | **0.706** | **0.701** | **0.701** | **0.704** | **0.706** | 0.635 | 0.636 | 0.591 | 0.590 |
| | image noise | 0.490 | 0.492 | 0.493 | **0.680** | **0.655** | 0.641 | 0.649 | 0.626 | 0.632 | 0.598 | 0.516 | 0.515 | 0.507 | 0.507 |
| $\alpha=5$ | text noise | 0.621 | 0.609 | 0.601 | **0.692** | **0.688** | **0.696** | **0.693** | 0.683 | **0.694** | **0.693** | 0.638 | 0.638 | 0.592 | 0.591 |
| | audio noise | 0.605 | 0.594 | 0.587 | **0.699** | **0.700** | **0.703** | **0.701** | **0.697** | **0.700** | **0.700** | 0.629 | 0.631 | 0.584 | 0.582 |
| | image missing | 0.554 | 0.550 | 0.545 | 0.637 | 0.652 | **0.658** | 0.654 | 0.651 | 0.651 | 0.637 | 0.546 | 0.547 | 0.540 | 0.539 |
| | text missing | 0.606 | 0.596 | 0.587 | 0.630 | 0.631 | **0.650** | 0.645 | 0.638 | **0.667** | **0.665** | 0.614 | 0.620 | 0.583 | 0.583 |
| | audio missing | 0.608 | 0.599 | 0.587 | **0.681** | 0.670 | 0.676 | **0.678** | 0.674 | 0.668 | 0.650 | 0.601 | 0.603 | 0.593 | 0.593 |
| | normal | 0.586 | 0.594 | 0.616 | 0.595 | **0.702** | **0.702** | **0.702** | **0.704** | **0.704** | 0.638 | 0.631 | 0.597 | 0.591 | 0.590 |
| | image noise | 0.487 | 0.488 | 0.490 | 0.496 | 0.649 | 0.626 | 0.647 | 0.621 | 0.621 | 0.502 | 0.515 | 0.508 | 0.506 | 0.506 |
| $\alpha=10$ | text noise | 0.592 | 0.599 | 0.624 | 0.601 | 0.686 | 0.685 | **0.694** | 0.683 | **0.692** | 0.644 | 0.634 | 0.597 | 0.591 | 0.591 |
| | audio noise | 0.578 | 0.586 | 0.605 | 0.587 | **0.703** | **0.697** | **0.699** | **0.699** | **0.702** | 0.623 | 0.632 | 0.592 | 0.582 | 0.582 |
| | image missing | 0.533 | 0.541 | 0.547 | 0.541 | 0.651 | **0.656** | **0.664** | 0.649 | 0.647 | 0.553 | 0.545 | 0.542 | 0.539 | 0.539 |
| | text missing | 0.581 | 0.585 | 0.604 | 0.588 | 0.626 | 0.634 | **0.649** | **0.644** | **0.659** | 0.609 | 0.620 | 0.588 | 0.583 | 0.583 |
| | audio missing | 0.582 | 0.588 | 0.606 | 0.586 | 0.665 | 0.671 | **0.682** | 0.668 | 0.658 | 0.631 | 0.602 | 0.593 | 0.593 | 0.593 |

Table 12: Results of different $\alpha$ and $\beta$ on MOSEI dataset under late fusion. Bold indicates better or equal performance than MT-early, and red font is the weight we select.

| | test case | $\beta = 0$ | $\beta = 0.1$ | $\beta = 0.2$ | $\beta = 0.4$ | $\beta = 0.6$ | $\beta = 0.8$ | $\beta = 1$ | $\beta = 1.2$ | $\beta = 1.4$ | $\beta = 1.6$ | $\beta = 1.8$ | $\beta = 2$ | $\beta = 4$ | $\beta = 10$ |
|---|---|---|---|---|---|---|---|---|---|---|---|---|---|---|---|
| | normal | 0.748 | **0.624** | 0.652 | 0.780 | 0.702 | **0.617** | **0.628** | 0.777 | 0.724 | **0.634** | **0.621** | 0.778 | 0.740 | **0.620** |
| | image noise | 0.775 | **0.642** | 0.673 | 0.874 | 0.718 | **0.649** | **0.663** | 0.876 | 0.764 | **0.655** | **0.663** | 0.807 | 0.819 | 0.715 |
| $\alpha = 0.1$ | text noise | 0.811 | **0.823** | 0.814 | 0.843 | **0.819** | 0.840 | **0.810** | 0.849 | **0.820** | 0.831 | **0.819** | 0.812 | 1.007 | **0.809** |
| | audio noise | 0.751 | **0.627** | 0.655 | 0.775 | 0.704 | **0.618** | **0.633** | 0.772 | 0.723 | **0.640** | **0.623** | 0.780 | 0.739 | **0.622** |
| | image missing | 0.748 | **0.626** | 0.658 | 0.797 | 0.708 | **0.619** | **0.632** | 0.797 | 0.728 | 0.639 | **0.629** | 0.781 | 0.741 | **0.632** |
| | text missing | **0.812** | **0.814** | **0.812** | 0.848 | **0.820** | 0.832 | **0.810** | 0.856 | 0.825 | 0.827 | **0.817** | **0.813** | 1.011 | **0.810** |
| | audio missing | 0.753 | **0.625** | 0.651 | 0.796 | 0.707 | **0.620** | **0.627** | 0.795 | 0.725 | **0.632** | **0.623** | 0.773 | 0.742 | **0.624** |
| | normal | 0.653 | **0.630** | 0.793 | 0.781 | 0.721 | 0.730 | 0.712 | **0.629** | **0.630** | 0.706 | 0.645 | **0.617** | **0.625** | **0.628** |
| | image noise | 0.686 | **0.659** | 0.855 | 0.875 | 0.727 | 0.742 | 0.746 | 0.668 | 0.686 | 0.724 | 0.688 | **0.659** | 0.667 | 0.771 |
| $\alpha = 0.2$ | text noise | 0.825 | 0.818 | 0.864 | 0.837 | 0.852 | 0.831 | 0.884 | **0.824** | 0.818 | 0.816 | 0.807 | 0.827 | **0.821** | 0.817 |
| | audio noise | 0.655 | **0.635** | 0.793 | 0.777 | 0.720 | 0.732 | 0.710 | **0.632** | **0.634** | 0.707 | 0.649 | **0.618** | **0.625** | **0.632** |
| | image missing | 0.661 | **0.635** | 0.797 | 0.800 | 0.719 | 0.736 | 0.711 | **0.633** | **0.627** | 0.708 | 0.656 | **0.622** | **0.627** | **0.628** |
| | text missing | 0.826 | **0.817** | 0.865 | 0.843 | 0.864 | 0.840 | 0.903 | **0.818** | **0.817** | 0.819 | 0.808 | 0.814 | **0.817** | 0.816 |
| | audio missing | 0.656 | **0.629** | 0.808 | 0.799 | 0.727 | 0.729 | 0.718 | **0.637** | **0.632** | 0.711 | 0.643 | **0.622** | **0.628** | **0.628** |
| | normal | **0.626** | 0.762 | **0.635** | 0.670 | 0.640 | 0.640 | **0.619** | 0.629 | 0.608 | **0.624** | **0.625** | 0.643 | **0.635** | 0.781 |
| | image noise | 0.675 | 0.786 | **0.663** | 0.684 | 0.672 | **0.660** | 0.673 | 0.667 | 0.643 | 0.665 | **0.658** | 0.666 | **0.655** | 0.815 |
| $\alpha = 1$ | text noise | 0.816 | **0.823** | 0.807 | 0.814 | **0.819** | 0.835 | 0.815 | 0.815 | 0.807 | 0.818 | 0.822 | 0.829 | **0.820** | 0.808 |
| | audio noise | **0.629** | 0.764 | **0.637** | 0.674 | **0.641** | 0.644 | **0.622** | 0.631 | 0.615 | **0.623** | **0.630** | 0.647 | **0.640** | 0.781 |
| | image missing | **0.626** | 0.765 | 0.640 | 0.672 | 0.643 | 0.639 | **0.623** | 0.629 | 0.612 | **0.629** | **0.625** | 0.645 | **0.635** | 0.789 |
| | text missing | 0.816 | 0.826 | 0.807 | 0.814 | 0.817 | 0.828 | 0.814 | 0.809 | 0.801 | 0.812 | 0.820 | 0.825 | 0.819 | 0.809 |
| | audio missing | **0.633** | 0.759 | **0.638** | 0.671 | 0.642 | 0.642 | **0.623** | 0.638 | 0.609 | 0.627 | **0.628** | 0.642 | **0.631** | 0.790 |
| | normal | 0.775 | 0.832 | **0.621** | **0.625** | **0.621** | **0.625** | 0.708 | 0.643 | 0.720 | **0.635** | 0.707 | 0.718 | 0.790 | 0.810 |
| | image noise | 0.779 | 0.852 | 0.670 | **0.662** | 0.677 | **0.648** | 0.732 | 0.678 | 0.816 | 0.666 | 0.721 | 0.754 | 0.842 | 0.832 |
| $\alpha = 5$ | text noise | 0.849 | 0.833 | **0.810** | 0.821 | 0.813 | 0.814 | 0.812 | 0.852 | 1.006 | **0.815** | **0.819** | 0.806 | 0.809 | 0.811 |
| | audio noise | 0.777 | 0.833 | **0.626** | **0.627** | **0.621** | **0.628** | 0.711 | 0.645 | 0.723 | **0.641** | 0.714 | 0.720 | 0.792 | 0.812 |
| | image missing | 0.774 | 0.835 | **0.627** | **0.635** | **0.634** | **0.631** | 0.718 | 0.646 | 0.721 | 0.642 | 0.708 | 0.718 | 0.798 | 0.820 |
| | text missing | 0.851 | 0.833 | **0.811** | 0.821 | **0.811** | **0.813** | **0.811** | 0.849 | 1.013 | **0.812** | **0.817** | 0.809 | 0.809 | 0.811 |
| | audio missing | 0.782 | 0.862 | **0.621** | **0.628** | **0.632** | **0.626** | 0.713 | 0.646 | 0.728 | **0.632** | 0.708 | 0.720 | 0.794 | 0.814 |
| | normal | 0.840 | 0.841 | 0.628 | 0.731 | 0.645 | 0.742 | **0.628** | 0.643 | **0.629** | 0.640 | **0.616** | 0.712 | 0.746 | 0.817 |
| | image noise | 0.841 | 0.840 | 0.672 | 0.749 | 0.670 | 0.789 | **0.647** | 0.668 | 0.675 | 0.672 | 0.671 | 0.742 | 0.879 | 0.834 |
| $\alpha = 10$ | text noise | 0.839 | 0.840 | **0.809** | 0.826 | **0.811** | 0.868 | 0.829 | 0.828 | **0.813** | **0.815** | **0.812** | 0.806 | **0.813** | 0.817 |
| | audio noise | 0.840 | 0.842 | **0.631** | 0.736 | 0.646 | 0.745 | **0.631** | 0.648 | **0.635** | 0.647 | **0.618** | 0.713 | 0.744 | 0.819 |
| | image missing | 0.855 | 0.856 | **0.632** | 0.738 | 0.647 | 0.742 | **0.626** | 0.642 | **0.631** | **0.637** | **0.621** | 0.716 | 0.760 | 0.825 |
| | text missing | 0.839 | 0.840 | **0.811** | 0.838 | **0.814** | 0.880 | **0.820** | 0.824 | **0.812** | **0.814** | **0.811** | 0.809 | 0.818 | 0.817 |
| | audio missing | 0.842 | 0.846 | **0.631** | 0.731 | 0.647 | 0.750 | **0.629** | 0.643 | **0.629** | 0.640 | **0.618** | 0.716 | 0.749 | 0.820 |

Table 13: Results of our proposed method with minimizing OT distance on all datasets under early fusion. The bold numbers mean the best performance. The bigger AUROC and F1 and smaller MAE refer to better performance.

| | | | normal test | test with modality noise | | | test with missing modality | | | |
|---|---|---|---|---|---|---|---|---|---|---|
| Datasets | Metric | Methods | normal | image/video noise | text noise | audio noise | image/video missing | text missing | audio missing | average |
| Hateful Memes | AUROC↑ | minimize OT | 0.724 | 0.574 | 0.674 | - | 0.621 | 0.658 | - | 0.650 |
| | | WMA | **0.748** | **0.583** | **0.682** | - | **0.645** | **0.659** | - | **0.663** |
| MM-IMDb | F1↑ | minimize OT | 0.551 | 0.309 | 0.462 | - | 0.453 | 0.415 | - | 0.438 |
| | | WMA | **0.563** | **0.355** | **0.471** | - | **0.477** | **0.437** | - | **0.461** |
| UR-FUNNY | AUROC↑ | minimize OT | 0.705 | 0.661 | 0.679 | 0.633 | **0.628** | 0.658 | 0.699 | 0.666 |
| | | WMA | **0.708** | **0.681** | **0.686** | **0.643** | 0.624 | **0.674** | **0.703** | **0.674** |
| MOSEI | MAE↓ | minimize OT | 0.624 | 0.711 | 0.855 | 0.676 | 0.629 | 0.831 | **0.623** | 0.707 |
| | | WMA | **0.615** | **0.676** | **0.824** | **0.664** | **0.619** | **0.813** | **0.623** | **0.691** |

Table 14: Results of our proposed method with different distance metrics on all datasets under late fusion. The bold numbers mean the best performance. The bigger AUROC and F1 and smaller MAE refer to better performance.

| | | | normal test | test with modality noise | | | test with missing modality | | | |
|---|---|---|---|---|---|---|---|---|---|---|
| Datasets | Metric | Methods | normal | image/video noise | text noise | audio noise | image/video missing | text missing | audio missing | average |
| Hateful Memes | AUROC↑ | minimize OT | 0.719 | 0.664 | 0.666 | - | 0.678 | 0.632 | - | 0.672 |
| | | WMA | **0.728** | **0.671** | **0.667** | - | **0.692** | **0.663** | - | **0.684** |
| MM-IMDb | F1↑ | minimize OT | 0.609 | 0.463 | 0.506 | - | 0.554 | 0.415 | - | 0.509 |
| | | WMA | **0.610** | **0.489** | **0.510** | - | **0.559** | **0.426** | - | **0.519** |
| UR-FUNNY | AUROC↑ | minimize OT | 0.604 | 0.492 | 0.611 | 0.595 | 0.542 | 0.588 | 0.594 | 0.575 |
| | | WMA | **0.710** | **0.660** | **0.698** | **0.702** | **0.660** | **0.672** | **0.683** | **0.684** |
| MOSEI | MAE↓ | minimize OT | 0.626 | 0.675 | 0.816 | 0.629 | 0.626 | 0.816 | 0.633 | 0.689 |
| | | WMA | **0.608** | **0.643** | **0.807** | **0.615** | **0.612** | **0.801** | **0.609** | **0.671** |

Table 15: Results of our proposed method with different distance metrics on all datasets and test cases for MT-early. The bold numbers mean the best performance. The bigger AUROC and F1 and smaller MAE refer to better performance.

| Datasets | Metric | Methods | normal test | test with modality noise | | | test with missing modality | | | |
|---|---|---|---|---|---|---|---|---|---|---|
| | | | normal | image/video noise | text noise | audio noise | image/video missing | text missing | audio missing | average |
| Hateful Memes | AUROC↑ | MT-early | 0.730 | 0.547 | 0.666 | - | 0.626 | 0.653 | - | 0.644 |
| | | MT-early-CS | 0.742 | 0.575 | 0.674 | - | 0.656 | **0.672** | - | **0.664** |
| | | MT-early-JS | 0.743 | 0.560 | 0.673 | - | 0.635 | 0.668 | - | 0.656 |
| | | MT-early-TV | **0.750** | 0.563 | 0.680 | - | 0.615 | 0.661 | - | 0.654 |
| | | MT-early-WMA | 0.748 | **0.583** | **0.682** | - | **0.645** | 0.659 | - | 0.663 |
| MM-IMDb | F1↑ | MT-early | 0.551 | 0.251 | 0.462 | - | 0.464 | 0.412 | - | 0.428 |
| | | MT-early-CS | 0.559 | 0.342 | 0.464 | - | 0.473 | 0.417 | - | 0.451 |
| | | MT-early-JS | 0.558 | 0.336 | 0.470 | - | 0.476 | 0.417 | - | 0.451 |
| | | MT-early-TV | **0.565** | 0.308 | 0.462 | - | **0.480** | 0.414 | - | 0.446 |
| | | MT-early-WMA | 0.563 | **0.355** | **0.471** | - | 0.477 | **0.437** | - | **0.461** |
| UR-FUNNY | AUROC↑ | MT-early | 0.700 | 0.645 | 0.678 | 0.635 | 0.612 | 0.670 | 0.696 | 0.662 |
| | | MT-early-CS | 0.711 | 0.656 | 0.682 | 0.624 | 0.617 | 0.679 | **0.708** | 0.668 |
| | | MT-early-JS | 0.711 | 0.660 | 0.664 | 0.639 | 0.594 | 0.663 | 0.705 | 0.662 |
| | | MT-early-TV | 0.708 | 0.665 | 0.685 | 0.630 | 0.622 | **0.680** | 0.704 | 0.671 |
| | | MT-early-WMA | **0.712** | **0.681** | **0.686** | **0.643** | **0.624** | 0.674 | 0.703 | **0.675** |
| MOSEI | MAE↓ | MT-early | 0.633 | 0.691 | 0.834 | 0.672 | 0.651 | 0.826 | 0.635 | 0.706 |
| | | MT-early-CS | 0.624 | 0.677 | 0.825 | 0.671 | 0.631 | 0.821 | 0.633 | 0.697 |
| | | MT-early-JS | 0.617 | 0.680 | 0.822 | 0.665 | 0.642 | 0.814 | 0.632 | 0.696 |
| | | MT-early-TV | **0.612** | 0.723 | **0.821** | 0.668 | 0.632 | 0.822 | 0.635 | 0.702 |
| | | MT-early-WMA | 0.615 | **0.676** | 0.824 | **0.664** | **0.619** | **0.813** | **0.623** | **0.691** |

Table 16: Results of our proposed method with different distance metrics on all datasets and test cases for MT-late. The bold numbers mean the best performance. The bigger AUROC and F1 and smaller MAE refer to better performance.

| Datasets | Metric | Methods | normal test | test with modality noise | | | test with missing modality | | | |
|---|---|---|---|---|---|---|---|---|---|---|
| | | | normal | image/video noise | text noise | audio noise | image/video missing | text missing | audio missing | average |
| Hateful Memes | AUROC↑ | MT-late | 0.718 | 0.652 | 0.666 | - | 0.680 | 0.622 | - | 0.668 |
| | | MT-late-CS | **0.728** | **0.675** | **0.679** | - | 0.687 | 0.621 | - | 0.678 |
| | | MT-late-JS | 0.724 | 0.666 | 0.674 | - | 0.664 | 0.639 | - | 0.673 |
| | | MT-late-TV | 0.724 | 0.665 | 0.671 | - | 0.680 | 0.633 | - | 0.675 |
| | | MT-late-WMA | **0.728** | 0.671 | 0.667 | - | **0.692** | **0.663** | - | **0.684** |
| MM-IMDb | F1↑ | MT-late | 0.602 | 0.469 | 0.488 | - | 0.553 | 0.414 | - | 0.505 |
| | | MT-late-CS | **0.612** | 0.459 | 0.478 | - | 0.555 | 0.418 | - | 0.504 |
| | | MT-late-JS | 0.608 | 0.475 | 0.500 | - | 0.550 | 0.421 | - | 0.511 |
| | | MT-late-TV | 0.609 | 0.470 | 0.490 | - | 0.549 | 0.419 | - | 0.507 |
| | | MT-late-WMA | 0.610 | **0.489** | **0.510** | - | **0.559** | **0.426** | - | **0.519** |
| UR-FUNNY | AUROC↑ | MT-late | 0.700 | 0.651 | 0.686 | 0.693 | 0.654 | 0.641 | 0.676 | 0.672 |
| | | MT-late-CS | 0.702 | 0.661 | 0.695 | 0.690 | 0.655 | 0.650 | 0.680 | 0.676 |
| | | MT-late-JS | **0.710** | 0.654 | 0.692 | 0.698 | 0.640 | 0.662 | 0.677 | 0.676 |
| | | MT-late-TV | 0.708 | 0.654 | 0.693 | 0.691 | 0.651 | 0.656 | 0.678 | 0.676 |
| | | MT-late-WMA | **0.710** | 0.660 | **0.698** | **0.702** | 0.660 | **0.672** | **0.683** | **0.684** |
| MOSEI | MAE↓ | MT-late | 0.638 | 0.665 | 0.826 | 0.642 | 0.638 | 0.822 | 0.640 | 0.696 |
| | | MT-late-CS | 0.612 | 0.658 | 0.822 | **0.614** | 0.621 | 0.821 | 0.616 | 0.681 |
| | | MT-late-JS | 0.613 | 0.703 | 0.811 | 0.615 | 0.620 | 0.814 | 0.618 | 0.685 |
| | | MT-late-TV | 0.613 | 0.645 | 0.809 | 0.615 | 0.619 | 0.809 | 0.620 | 0.676 |
| | | MT-late-WMA | **0.608** | **0.643** | **0.807** | 0.615 | **0.612** | **0.801** | **0.609** | **0.671** |

