# OpenReview forum: "Wasserstein Modality Alignment Makes Your Multimodal Transformer More Robust"
_TMLR — Accepted by TMLR_

### Review · Reviewer_TJU2 · 2024-09-09

**Summary Of Contributions:**

The paper aims to address the multi-modal fusion problem in the multi-modal transformer architecture when processing multi-modal data.

The paper mainly addresses the challenge that prior multi-modal fusion directly fuses all the modalities without alignment, where the final prediction may only exploit the easiest modality for prediction. The paper shows the contributions of different input modalities to demonstrate this claim.

The paper proposes an optimal-transport-based Wasserstein Modality Alignment (WMA) loss to align features of different modalities. The loss minimizes the difference between the optimal-transport distance of different modalities and a reference distance.

The paper conducts experiments on four datasets including modalities text, image, audio and video to verify the effectiveness of the ideas.

**Audience:**

Yes

**Claims And Evidence:**

Yes

**Requested Changes:**

Could the authors address the problem in the weakness?

**Strengths And Weaknesses:**

Strength:

The fusion of multi-modal data is an important next step for foundation models. The paper points out an important problem of current multi-modal transformer architectures for multi-modal fusion, which is that some modality dominates the prediction and other modalities are actually not used. This is because a common learning paradigm tends to perform lazy and greedy learning and learn the easiest way to solve the problem.

The paper plots the contributions of different modalities to the final prediction, which shows that the contribution distribution is very skewed. This further verifies the claim in the challenge.

The proposed modality alignment loss is a plug-and-play method, which could be used in varieties of transformer architectures.

Experimental results show that using the proposed WMA method improves the multi-modal prediction performance. Also, the authors compare the performance between directly minimizing the optimal-transport distance and minimizing the difference to a reference distance. The results support the claim that minimizing the difference to a reference distance is a better choice.


Weakness:

Although empirically effective, the reason why aligning the features of different modalities could address the problem of skewed contribution of modalities. This is obviously a regularization loss but how it functions still needs some explanation.

The reference distance is decided as the average of the first batch. This is a quite random choice, which is severely influenced by the random seed. Could the authors demonstrate that the performance is robust to the choice of the batch of data?

---

> ### Author Response · Authors · 2024-09-29
>
> $$Weakness1$$Thanks for your valuable comments. The reason why WMA works is that it aligns the feature distributions of different modalities so that the learning does not only rely on a single modality that leads to the shortcut learning, which makes the model vulnerable to distribution shifts. Section 3.2 explains how the regularization works by comparing a model trained with WMA and without. We are currently conducting some theoretical analysis and will provide an update if it is completed.
> $$Weakness2$$ Thanks for your valuable comments. We are running experiments with different random seeds and will update as they are fully completed. The current result suggests that the reference distance is stable when using different batches of data.

---

> ### Author Response · Authors · 2024-10-06
>
> **Weakness2**
>
> The WMA algorithm uses the average OT value of the first batch of samples as the reference. To verify the stability of the method under different random seeds, we conduct an ablation study using the UR-FUNNY dataset with early fusion. For each experiment, we use 10 different random seeds: [0, 10, 20, 30, 40, 50, 60, 70, 80, 90]. We report the average value along with the standard deviation across all baselines in Table below.  From the table, we observe that the results of all methods exhibit slight variations under different random seeds. However, the proposed WMA method consistently demonstrates a clear advantage in terms of both performance and robustness. Additionally, we find that the optimal values for $\alpha$ and $\beta$ in the WMA method vary slightly depending on the seed (e.g., [$\alpha$=5, $\beta$=1.8] for random seed = 0, and [$\alpha$=1, $\beta$=1.2] for random seed = 20). This adaptability highlights the WMA method's strength in achieving optimal modality alignment under varying conditions.  We have updated the paper and highlighted the change in red.
>
> *Table: The performance of different methods on UR-FUNNY dataset under early fusion with different random seeds.*
>
> | Models            | Normal          | Video Noise     | Text Noise      | Audio Noise     | Video Missing   | Text Missing    | Audio Missing   |
> |-------------------|-----------------|-----------------|-----------------|-----------------|-----------------|-----------------|-----------------|
> | MT-early          | 0.698 ± 0.003   | 0.642 ± 0.004   | 0.677 ± 0.003   | 0.634 ± 0.004   | 0.615 ± 0.006   | 0.673 ± 0.003   | 0.691 ± 0.006   |
> | MT-early-GWMAC    | 0.701 ± 0.005   | 0.653 ± 0.004   | 0.675 ± 0.004   | 0.628 ± 0.006   | 0.610 ± 0.008   | 0.671 ± 0.004   | 0.700 ± 0.003   |
> | MT-early-RegBN    | **0.716** ± 0.004   | 0.613 ± 0.009   | 0.663 ± 0.005   | 0.617 ± 0.011   | 0.613 ± 0.003   | 0.648 ± 0.002   | 0.696 ± 0.004   |
> | MT-early-MIB      | 0.710 ± 0.003   | 0.643 ± 0.008   | 0.682 ± 0.006   | 0.632 ± 0.005   | 0.611 ± 0.003   | 0.666 ± 0.003   | 0.692 ± 0.003   |
> | MT-early-WMA      | 0.713 ± 0.003   | **0.681** ± 0.004   | **0.688** ± 0.004   | **0.646** ± 0.004   | **0.626** ± 0.003   | **0.675** ± 0.004   | **0.708** ± 0.004   |

---

### Review · Reviewer_B5qq · 2024-09-14

**Summary Of Contributions:**

The authors propose WMA, an implicit regularization method to align the Wasserstein distance between different modalities within a multimodal transformer. This method does not introduce any additional trainable parameters. Instead of directly minimizing the optimal transport (OT) distance between modalities, WMA aligns modalities based on the task requirements. This is achieved through two hyperparameters, allowing for a search for the optimal alignment. The paper provides insights into the degree of modality alignment required for different tasks, showing that task-dependent alignment is more effective than simply minimizing the OT distance.

**Audience:**

Yes

**Broader Impact Concerns:**

Multimodal systems can sometimes amplify biases present in the training data. If WMA improves the performance of such systems, it might also enhance the impact of any biases, leading to unfair or discriminatory outcomes in applications like surveillance, hiring, or credit scoring. The use of implicit regularization through WMA might make the model's decision process less transparent. This could be a concern in high-stakes applications where explainability is crucial.

**Claims And Evidence:**

Yes

**Requested Changes:**

See weakness.

**Strengths And Weaknesses:**

Strengths:

The proposal of WMA as a novel method for implicit regularization in multimodal transformers is innovative and addresses a significant problem in the field. The concept of task-dependent modality alignment is a strong contribution, offering flexibility and adaptability that is crucial for real-world applications.

Weaknesses:

1. While the empirical results are strong, the paper could benefit from a deeper theoretical analysis of why WMA works and under what conditions it is most effective.

2. The paper focuses on a limited set of modalities (text, image, audio). It would be beneficial to understand how WMA generalizes to other types of modalities or data.

3. The paper does not discuss the computational overhead introduced by WMA. Analyzing the efficiency and scalability, especially for larger models or datasets, would be important.

4. While the paper compares WMA with several baselines, it could strengthen its claims by comparing against the latest state-of-the-art methods in multimodal learning.

---

> ### Author Response · Authors · 2024-09-29
>
> $$Weakness1$$Thanks for your valuable comments. We agree that a more in-depth theoretical analysis will provide readers with a deeper understanding of our proposed approach. However, our work is mainly driven by the empirical study and theoretical results are not stated as our major contribution. The reason why WMA works is that it aligns the feature distributions of different modalities so that the learning does not only rely on a single modality that leads to the shortcut learning, which makes the model vulnerable to distribution shifts. Thus, our WMA is most effective when there are out-of-distribution data during inference as shown in our experiment. We are currently conducting some theoretical analysis and will provide an update if it is completed.
> $$Weakness2$$ Thanks for your valuable comments. In fact, the dataset we use includes four modalities. Image, Text, Audio, and Video. To verify the ability of our proposed method to generalize over the other modalities, we chose a medical dataset, medfuse-I (this dataset contains EHR data and X-ray images). We are running the experiments and will update the results as they are completed.
> $$Weakness3$$ Thanks for your valuable comments. We agree that it is critical to consider computational efficiency in this process. We use a fast OT solver during training that does not induce a substantial computational overhead. Moreover, the OT distance is only computed during the training phase instead of during inference, indicating that the computational cost induced by OT is amortized. We are running the experiment and  will add the computational time of different methods in the updated version.
> $$Weakness4$$ Thanks for your valuable comments. We have incorporated the recent strong baseline RegBN Ghahremani Boozandani & Wachinger (2024). We did not choose those well-known multimodal models, e.g. llava, gpt4o because we mainly focus on studying the discriminative multimodal models rather than LLM-based multimodal models. We believe the discriminative multimodal models are still meaningful today as they are better at specific tasks and more cost-efficient than LLM-based models.

---

> ### Author Response · Authors · 2024-10-06
>
> **Weakness2**
>
> We have added  a medical dataset, MedFuse-In-hospital mortality (MedFuse-I): This real-world dataset contains EHR and X-ray data for each patient. This binary classification task aims to predict in-hospital mortality after the first 48 hours spent in the ICU. The EHR is time-series data with 17 clinical variables, among which five are categorical and 12 are continuous. Each EHR is paired with the last chest X-ray image collected during the ICU stay. The numbers of the samples in the training/val/testing dataset are 18845, 2138 and 5243. We use AUROC as the metric.
> Robustness test setting: This real-world dataset itself lacks X-ray images for 74% of samples. We do not perform additional missing value settings to evaluate all methods faithfully. For the modal noise setting we are consistent with the Section 4.1. The performance of all methods on MedFuse-I is shown below. From the tables we observe the same trend with Section 4.1 that the proposed WMA method beats all baselines in terms of both performance and robustness. We believe that the proposed WMA method generalizes well on different tasks and different types of datasets. We have updated the paper and highlighted the change in red.
>
> *Table: Performance of all methods on Medfuse-I dataset under early fusion and late fusion*
>
> | Models           | Normal Test/Image Missed (Early Fusion) | EHR Noise (Early Fusion) | Image Noise (Early Fusion) | Normal Test/Image Missed (Late Fusion) | EHR Noise (Late Fusion) | Image Noise (Late Fusion) |
> |------------------|-----------------------------------------|---------------------------|----------------------------|----------------------------------------|-------------------------|---------------------------|
> | MT        | 0.840 | 0.723 | 0.762 | 0.848    | 0.742                   | 0.781                     |
> | MT-GWMAC   | 0.844| 0.749  | 0.787        | 0.853                             | 0.758                   | 0.794                     |
> | MT-RegBN   | 0.855  | 0.722     | 0.751                      | 0.859                                  | 0.743                   | 0.785                     |
> | MT-MIB     | 0.852| 0.715       | 0.744                      | 0.860                                  | 0.738                   | 0.774                     |
> | MT-WMA     | **0.861**  | **0.755**       | **0.796**                     | **0.864**                                  | **0.775**                   | **0.813**                     |
>
>
> **Weakness3**
>
> We compare the training times of all methods  on the UR-FUNNY and MOSEI datasets under early fusion and late fusion. Specifically, we report the time required for a training step, including both the forward and backpropagation processes for a single batch of data. As shown in Table below, our method surpasses the strong baselines GWMAC, RegBN, and MIB in terms of training speed. This improvement can be attributed to the fact that GWMAC, RegBN, and MIB introduce additional parameters, while our WMA method operates without training any new parameters. Although the fast OT method  used in the paper increases the computation time for distance calculations, the overall impact on runtime remains minimal. During inference, our method retains the same number of parameters and execution speed as the basic MT. Therefore, the proposed WMA method has a better computational efficiency compared with the recent baselines. We have updated the paper and highlighted the change in red.
>
> *Table: Computational time of a step for different methods on UR-FUNNY and MOSEI datasets under early fusion and late fusion.**
> | Models/Datasets  | Early Fusion (UR-FUNNY) | Early Fusion (MOSEI) | Late Fusion (UR-FUNNY) | Late Fusion (MOSEI) |
> |------------------|-------------------------|-----------------------|------------------------|----------------------|
> | MT          | 2.364s                  | 2.288s                | 1.735s                 | 1.506s               |
> | MT-GWMAC    | 3.290s                  | 2.801s                | 2.172s                 | 2.054s               |
> | MT-RegBN    | 3.962s                  | 3.287s                | 3.298s                 | 2.990s               |
> | MT-MIB      | 3.476s                  | 2.947s                | 2.786s                 | 2.409s               |
> | **MT-WMA**      | **2.788s**                  | **2.570s**                | **2.098s**                 | **1.920s**               |
> | MT-CS      | 2.381s                  | 2.312s                | 1.778s                 | 1.554s               |
> | MT-JS      | 2.398s                  | 2.472s                | 1.816s                 | 1.609s               |
> | MT-TV      | 2.409s                  | 2.335s                | 1.983s                 | 1.761s               |
> | | | | | |
>
>
> [1] Xie, Y., Wang, X., Wang, R. and Zha, H., 2020, August. A fast proximal point method for computing exact wasserstein distance. In Uncertainty in artificial intelligence (pp. 433-453). PMLR.

---

### Review · Reviewer_k9gC · 2024-09-23

**Summary Of Contributions:**

The paper introduces Wasserstein Modality Alignment (WMA), a method to improve multimodal transformer models' robustness and performance by aligning modality representations using the optimal transport distance. WMA has the following advantages:

1. This alignment does not add trainable parameters, making it lightweight and efficient.

2. The approach regularizes the transformer during the fine-tuning stage to maintain an optimal modality distance. WMA adapts the alignment based on task requirements, rather than minimizing the distance between modalities outright.

3. The proposed method works for both early and late fusion settings in multimodal learning.

**Audience:**

Yes

**Claims And Evidence:**

Yes

**Requested Changes:**

Please refer to the **Weakness**.

**Strengths And Weaknesses:**

### Strength

1. As multimodal large foundation models are becoming popular, improving the performance of multimodal Transformers is an important topic. This paper is timely in addressing these challenges with a new and efficient alignment method.

2. The paper focuses on alignment during the fine-tuning stage, which is crucial since fine-tuning pre-trained models is widely used in practical applications. This scenario requires a more lightweight and efficient approach for multimodal learning compared to the resource-intensive pre-training phase.

### Weakness
1. The most recent dataset mentioned in Section 4.1 is from 2020. It would be beneficial to include newer datasets, such as LLaVA instruct, to justify the model's effectiveness on more current, large-scale benchmarks relevant to the evolving field of multimodal learning.

2. This paper does not discuss too much on the computational issue. Since the proposed method involves solving an OT problem,  it would be valuable to include an analysis of the additional time complexity or a real-time computation comparison with baseline methods.

---

> ### Author Response · Authors · 2024-09-29
>
> $$Weakness 1$$ Thanks for your valuable comments. This paper and all the baselines focus on the discriminative multimodal model instead of LLAVA-like generative multimodal models that rely on an LLM to generate. Therefore, we refer to the validation methodology employed in recently published works, which emphasizes validation across discriminative tasks [1, 2, 3]. We selected our dataset to align with this body of work. Furthermore, we use datasets from the popular benchmark: Multibench [4], strengthening the relevance and credibility of our approach. For generative multimodal models based on large language models, such as LLAVA, we acknowledge the importance of this emerging area and plan to explore it in future research.
>
> [1] Lee, Y.L., Tsai, Y.H., Chiu, W.C. and Lee, C.Y., 2023. Multimodal prompting with missing modalities for visual recognition. In Proceedings of the IEEE/CVF Conference on Computer Vision and Pattern Recognition (pp. 14943-14952).
>
> [2] Ghahremani Boozandani, M. and Wachinger, C., 2024. RegBN: Batch Normalization of Multimodal Data with Regularization. Advances in Neural Information Processing Systems, 36.
>
> [3] Liang, P.P., Cheng, Y., Fan, X., Ling, C.K., Nie, S., Chen, R., Deng, Z., Allen, N., Auerbach, R., Mahmood, F. and Salakhutdinov, R.R., 2024. Quantifying & modeling multimodal interactions: An information decomposition framework. Advances in Neural Information Processing Systems, 36.
>
> [4] Liang, P.P., Lyu, Y., Fan, X., Wu, Z., Cheng, Y., Wu, J., Chen, L., Wu, P., Lee, M.A., Zhu, Y. and Salakhutdinov, R., 2021. Multibench: Multiscale benchmarks for multimodal representation learning. Advances in neural information processing systems, 2021(DB1), p.1.
>
> $$Weakness 2$$ Thanks for your valuable comments. We agree that it is critical to consider computational efficiency in this process. We are running the experiment and will add the computational time of different methods in the updated version. We use a fast OT solver during training that does not induce a substantial computational overhead. Moreover, the OT distance is only computed during the training phase instead of during inference, indicating that the computational cost induced by OT is amortized.

---

> ### Author Response · Authors · 2024-10-06
>
> We compare the training times of all methods  on the UR-FUNNY and MOSEI datasets under early fusion and late fusion. Specifically, we report the time required for a training step, including both the forward and backpropagation processes for a single batch of data. As shown in Table below, our method surpasses the strong baselines GWMAC, RegBN, and MIB in terms of training speed. This improvement can be attributed to the fact that GWMAC, RegBN, and MIB introduce additional parameters, while our WMA method operates without training any new parameters. Although the fast OT method  used in the paper increases the computation time for distance calculations, the overall impact on runtime remains minimal. During inference, our method retains the same number of parameters and execution speed as the basic MT. Therefore, the proposed WMA method has a better computational efficiency compared with the recent baselines. We have updated the paper and highlighted the change in red.
>
> **Table: Computational time of a step for different methods on UR-FUNNY and MOSEI datasets under early fusion and late fusion.**
> | Models/Datasets  | Early Fusion (UR-FUNNY) | Early Fusion (MOSEI) | Late Fusion (UR-FUNNY) | Late Fusion (MOSEI) |
> |------------------|-------------------------|-----------------------|------------------------|----------------------|
> | MT          | 2.364s                  | 2.288s                | 1.735s                 | 1.506s               |
> | MT-GWMAC    | 3.290s                  | 2.801s                | 2.172s                 | 2.054s               |
> | MT-RegBN    | 3.962s                  | 3.287s                | 3.298s                 | 2.990s               |
> | MT-MIB      | 3.476s                  | 2.947s                | 2.786s                 | 2.409s               |
> | **MT-WMA**      | **2.788s**                  | **2.570s**                | **2.098s**                 | **1.920s**               |
> | MT-CS      | 2.381s                  | 2.312s                | 1.778s                 | 1.554s               |
> | MT-JS      | 2.398s                  | 2.472s                | 1.816s                 | 1.609s               |
> | MT-TV      | 2.409s                  | 2.335s                | 1.983s                 | 1.761s               |
> | | | | | |
>
>
>
>
> [1] Xie, Y., Wang, X., Wang, R. and Zha, H., 2020, August. A fast proximal point method for computing exact wasserstein distance. In Uncertainty in artificial intelligence (pp. 433-453). PMLR.

---

### Decision · Action_Editor_VSbk · 2024-12-19

**Recommendation:** Accept with minor revision

**Comment:**

Overall I believe that this is strong work that I recommend should be accepted, as did the two reviewers who provided their final recommendations.

There are only two small additions I would like to see for the minor revisions:
- A brief discussion on the relevance of the approach to generative tasks.  I think it is fine that the method is currently limited to discriminative tasks, but it needs to make this restriction clearer and it would be good to have a small bit of discussion about if and how it might be generalised to generative settings (or and explanation for why it can't if this is the case).
- As noted by one of the reviewers, multimodal systems can amplify biases in the data and its possible the approach could have impact (positive or negative) on this.  I think the paper would thus benefit from a brief discussion on broader impacts relating to how the approach might impact on bias and fairness.

**Audience:**

All reviewers believed that the paper will be of interest to the TMLR audience.  In particular, reviewers praised the work's strength in its novel method, empirical strength, and relevance to fine-tuning.  I think that it thus clearly meets the audience criteria.

**Claims And Evidence:**

This paper introduces Wasserstein Modality Alignment (WMA), a lightweight regularization method for improving the robustness and performance of multimodal transformers by aligning modality tokens using the Wasserstein distance during fine-tuning. Key benefits of the approach include not requiring additional trainable parameters and adapting the degree of alignment to task-specific requirements. Experimental results on multiple datasets across different fusion paradigms show improvement in both accuracy and robustness to noisy or missing modalities compared to baselines.

Initial concerns were raised by reviewers about limited dataset variability, computational overhead, and the robustness of the reference distribution.  However, the author's provided a significant rebuttal and update of the paper to address these, with all of the reviewers who followed up happy with the changes.  In particular, a number of new experiments were added that noticeably strengthened the work.

Concerns were also raised about the limited theoretical analysis and applicability to generative tasks.   While these were less addressed in the rebuttals, I do not think either would be a fair reason to reject the paper: the paper is upfront about it's empirical focus (and not all papers need theoretical results), while the use of transformers in discriminative settings is still widespread so this restriction does not mean in anyway that the work is not useful.

Overall, all reviewers indicated that they believe the work meets the claims and evidence criteria for TMLR and I agree with this assessment myself.